# Can Graph Neural Networks Count Substructures?

**Zhengdao Chen**
New York University
zc1216@nyu.edu

**Lei Chen**
New York University
lc3909@nyu.edu

**Soledad Villar**
Johns Hopkins University
soledad.villar@jhu.edu

**Joan Bruna**
New York University
bruna@cims.nyu.edu

## Abstract

The ability to detect and count certain substructures in graphs is important for solving many tasks on graph-structured data, especially in the contexts of computational chemistry and biology as well as social network analysis. Inspired by this, we propose to study the expressive power of graph neural networks (GNNs) via their ability to count attributed graph substructures, extending recent works that examine their power in graph isomorphism testing and function approximation. We distinguish between two types of substructure counting: induced-subgraph-count and subgraph-count, and establish both positive and negative answers for popular GNN architectures. Specifically, we prove that Message Passing Neural Networks (MPNNs), 2-Weisfeiler-Lehman (2-WL) and 2-Invariant Graph Networks (2-IGNs) cannot perform induced-subgraph-count of any connected substructure consisting of 3 or more nodes, while they can perform subgraph-count of star-shaped substructures. As an intermediary step, we prove that 2-WL and 2-IGNs are equivalent in distinguishing non-isomorphic graphs, partly answering an open problem raised in [38]. We also prove positive results for $k$-WL and $k$-IGNs as well as negative results for $k$-WL with a finite number of iterations. We then conduct experiments that support the theoretical results for MPNNs and 2-IGNs. Moreover, motivated by substructure counting and inspired by [45], we propose the Local Relational Pooling model and demonstrate that it is not only effective for substructure counting but also able to achieve competitive performance on molecular prediction tasks.

## 1 Introduction

In recent years, graph neural networks (GNNs) have achieved empirical success on processing data from various fields such as social networks, quantum chemistry, particle physics, knowledge graphs and combinatorial optimization [4, 5, 9, 10, 12, 13, 29, 31, 47, 55, 58, 66, 67, 69, 70, 71, 73, 74]. Thanks to such progress, there have been growing interests in studying the expressive power of GNNs. One line of work does so by studying their ability to distinguish non-isomorphic graphs. In this regard, Xu et al. [64] and Morris et al. [44] show that GNNs based on neighborhood-aggregation schemes are at most as powerful as the classical Weisfeiler-Lehman (WL) test [60] and propose GNN architectures that can achieve such level of power. While graph isomorphism testing is very interesting from a theoretical viewpoint, one may naturally wonder how relevant it is to real-world tasks on graph-structured data. Moreover, WL is powerful enough to distinguish almost all pairs of non-isomorphic graphs except for rare counterexamples [3]. Hence, from the viewpoint of graph isomorphism testing, existing GNNs are in some sense already not far from being maximally powerful, which could make the pursuit of more powerful GNNs appear unnecessary.

Another perspective is the ability of GNNs to approximate permutation-invariant functions on graphs. For instance, Maron et al. [41] and Keriven and Peyré [28] propose architectures that achieve universal approximation of permutation-invariant functions on graphs, though such models involve tensors with order growing in the size of the graph and are therefore impractical. Chen et al. [8] establishes an equivalence between the ability to distinguish any pair of non-isomorphic graphs and the ability to approximate arbitrary permutation-invariant functions on graphs. Nonetheless, for GNNs used in practice, which are not universally approximating, more efforts are needed to characterize *which* functions they can or cannot express. For example, Loukas [37] shows that GNNs under assumptions are Turing universal but lose power when their depth and width are limited, though the arguments rely on the nodes all having distinct features and the focus is on the asymptotic depth-width tradeoff. Concurrently to our work, Garg et al. [16] provide impossibility results of several classes of GNNs to decide graph properties including girth, circumference, diameter, radius, conjoint cycle, total number of cycles, and $k$-cliques. Despite these interesting results, we still need a perspective for understanding the expressive power of different classes of GNNs in a way that is intuitive, relevant to goals in practice, and potentially helpful in guiding the search for more powerful architectures.

Meanwhile, graph substructures (also referred to by various names including *graphlets*, *motifs*, *subgraphs* and *graph fragments*) are well-studied and relevant for graph-related tasks in computational chemistry [11, 13, 25, 26, 27, 46], computational biology [32] and social network studies [24]. In organic chemistry, for example, certain patterns of atoms called functional groups are usually considered indicative of the molecules' properties [33, 49]. In the literature of molecular chemistry, substructure counts have been used to generate molecular fingerprints [43, 48] and compute similarities between molecules [1, 53]. In addition, for general graphs, substructure counts have been used to create graph kernels [56] and compute spectral information [51]. The connection between GNNs and graph substructures is explored empirically by Ying et al. [68] to interpret the predictions made by GNNs. Thus, the ability of GNN architectures to count graph substructures not only serves as an intuitive theoretical measure of their expressive power but also is highly relevant to practical tasks.

In this work, we propose to understand the expressive power of GNN architectures via their ability to count attributed substructures, that is, counting the number of times a given pattern (with node and edge features) appears as a *subgraph* or *induced subgraph* in the graph. We formalize this question based on a rigorous framework, prove several results that partially answer the question for Message Passing Neural Networks (MPNNs) and Invariant Graph Networks (IGNs), and finally propose a new model inspired by substructure counting. In more detail, our main contributions are:

1. We prove that neither MPNNs [17] nor 2nd-order Invariant Graph Networks (2-IGNs) [41] can count *induced subgraphs* for any connected pattern of 3 or more nodes. For any such pattern, we prove this by constructing a pair of graphs that provably cannot be distinguished by any MPNN or 2-IGN but with different induced-subgraph-counts of the given pattern. This result points at an important class of simple-looking tasks that are provably hard for classical GNN architectures.

2. We prove that MPNNs and 2-IGNs can count *subgraphs* for star-shaped patterns, thus generalizing the results in Arvind et al. [2] to incorporate node and edge features. We also show that $k$-WL and $k$-IGNs can count *subgraphs* and *induced subgraphs* for patterns of size $k$, which provides an intuitive understanding of the hierarchy of $k$-WL's in terms of increasing power in counting substructures.

3. We prove that $T$ iterations of $k$-WL is unable to count *induced subgraphs* for *path* patterns of $(k + 1)2^T$ or more nodes. The result is relevant since real-life GNNs are often shallow, and also demonstrates an interplay between $k$ and depth.

4. Since substructures present themselves in local neighborhoods, we propose a novel GNN architecture called *Local Relation Pooling (LRP)*[1], with inspirations from Murphy et al. [45]. We empirically demonstrate that it can count both *subgraphs* and *induced subgraphs* on random synthetic graphs while also achieving competitive performances on molecular datasets. While variants of GNNs have been proposed to better utilize substructure information [34, 35, 42], often they rely on handcrafting rather than learning such information. By contrast, LRP is not only powerful enough to count substructures but also able to learn from data *which* substructures are relevant.

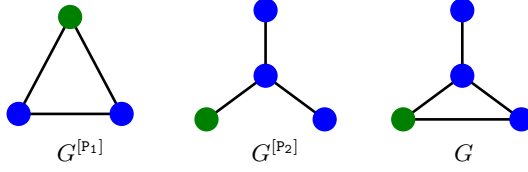

$G^{[\text{P}_1]}$       $G^{[\text{P}_2]}$       $G$

Figure 1: Illustration of the two types substructure-counts of the patterns $G^{[\text{P}_1]}$ and $G^{[\text{P}_2]}$ in the graph $G$, as defined in Section 2.1. The node features are indicated by colors. We have $\mathsf{C}_I(G; G^{[\text{P}_1]}) = \mathsf{C}_S(G; G^{[\text{P}_1]}) = 1$, and $\mathsf{C}_I(G; G^{[\text{P}_2]}) = 0$ while $\mathsf{C}_S(G; G^{[\text{P}_2]}) = 1$.

## 2 Framework

### 2.1 Attributed graphs, (induced) subgraphs and two types of counting

We define an *attributed graph* as $G = (V, E, x, e)$, where $V = [n] := \{1, ..., n\}$ is the set of vertices, $E \subset V \times V$ is the set of edges, $x_i \in \mathcal{X}$ represents the feature of node $i$, and $e_{i,j} \in \mathcal{Y}$ represent the feature of the edge $(i, j)$ if $(i, j) \in E$. The adjacency matrix $A \in \mathbb{R}^{n \times n}$is defined by $A_{i,j} = 1$ if $(i, j) \in E$ and 0 otherwise. We let $D_i = \sum_{j \in V} A_{i,j}$ denote the degree of node $i$. For simplicity, we only consider undirected graphs without self-connections or multi-edges. Note that an unattributed graph $G = (V, E)$ can be viewed as an attributed graph with identical node and edge features.

Unlike the node and edge features, the indices of the nodes are not inherent properties of the graph. Rather, different ways of ordering the nodes result in different representations of the same underlying graph. This is characterized by the definition of *graph isomorphism*: Two attributed graphs $G^{[1]} = (V^{[1]}, E^{[1]}, x^{[1]}, e^{[1]})$ and $G^{[2]} = (V^{[2]}, E^{[2]}, x^{[2]}, e^{[2]})$ are *isomorphic* if there exists a bijection $\pi : V^{[1]} \to V^{[2]}$ such that (1) $(i, j) \in E^{[1]}$ if and only if $(\pi(i), \pi(j)) \in E^{[2]}$, (2) $x_i^{[1]} = x_{\pi(i)}^{[2]}$ for all $i$ in $V^{[1]}$, and (3) $e_{i,j}^{[1]} = e_{\pi(i),\pi(j)}^{[2]}$ for all $(i, j) \in E^{[1]}$.

For $G = (V, E, x, e)$, a *subgraph* of $G$ is any graph $G^{[\text{S}]} = (V^{[\text{S}]}, E^{[\text{S}]}, x, e)$ with $V^{[\text{S}]} \subseteq V$ and $E^{[\text{S}]} \subseteq E$. An *induced subgraphs* of $G$ is any graph $G^{[\text{S}']} = (V^{[\text{S}']}, E^{[\text{S}']}, x, e)$ with $V^{[\text{S}']} \subseteq V$ and $E^{[\text{S}']} = E \cap (V^{[\text{S}']})^2$. In words, the edge set of an induced subgraph needs to include all edges in $E$ that have both end points belonging to $V^{[\text{S}']}$. Thus, an induced subgraph of $G$ is also its subgraph, but the converse is not true.

We now define two types of counting associated with *subgraphs* and *induced subgraphs*, as illustrated in Figure 1. Let $G^{[\text{P}]} = (V^{[\text{P}]}, E^{[\text{P}]}, x^{[\text{P}]}, e^{[\text{P}]})$ be a (typically smaller) graph that we refer to as a *pattern* or *substructure*. We define $\mathsf{C}_S(G, G^{[\text{P}]})$, called the *subgraph-count* of $G^{[\text{P}]}$ in $G$, to be the number of *subgraphs* of $G$ that are isomorphic to $G^{[\text{P}]}$. We define $\mathsf{C}_I(G; G^{[\text{P}]})$, called the *induced-subgraph-count* of $G^{[\text{P}]}$ in $G$, to be the number of *induced subgraphs* of $G$ that are isomorphic to $G^{[\text{P}]}$. Since all induced subgraphs are subgraphs, we always have $\mathsf{C}_I(G; G^{[\text{P}]}) \leq \mathsf{C}_S(G; G^{[\text{P}]})$.

Below, we formally define the ability for certain function classes to count substructures as the ability to distinguish graphs with different subgraph or induced-subgraph counts of a given substructure.

**Definition 2.1.** *Let $\mathcal{G}$ be a space of graphs, and $\mathcal{F}$ be a family of functions on $\mathcal{G}$. We say $\mathcal{F}$ is able to perform subgraph-count (or induced-subgraph-count) of a pattern $G^{[\text{P}]}$ on $\mathcal{G}$ if for all $G^{[1]}, G^{[2]} \in \mathcal{G}$ such that $\mathsf{C}_S(G^{[1]}, G^{[\text{P}]}) \neq \mathsf{C}_S(G^{[2]}, G^{[\text{P}]})$ (or $\mathsf{C}_I(G^{[1]}, G^{[\text{P}]}) \neq \mathsf{C}_I(G^{[2]}, G^{[\text{P}]})$), there exists $f \in \mathcal{F}$ that returns different outputs when applied to $G^{[1]}$ and $G^{[2]}$.*

In Appendix A, we prove an equivalence between Definition 2.1 and the notion of approximating *subgraph-count* and *induced-subgraph-count* functions on the graph space. Definition 2.1 also naturally allows us to define the ability of graph isomorphism tests to count substructures. A graph isomorphism test, such as the Weisfeiler-Lehman (WL) test, takes as input a pair of graphs and returns whether or not they are judged to be isomorphic. Typically, the test will return true if the two graphs are indeed isomorphic but does not necessarily return false for every pair of non-isomorphic graphs. Given such a graph isomorphism test, we say it is able to perform induced-subgraph-count (or subgraph-count) of a pattern $G^{[\text{P}]}$ on $\mathcal{G}$ if $\forall G^{[1]}, G^{[2]} \in \mathcal{G}$ such that $\mathsf{C}_I(G^{[1]}, G^{[\text{P}]}) \neq \mathsf{C}_I(G^{[2]}, G^{[\text{P}]})$ (or $\mathsf{C}_S(G^{[1]}, G^{[\text{P}]}) \neq \mathsf{C}_S(G^{[2]}, G^{[\text{P}]})$), the test can distinguish these two graphs.

# 3 Message Passing Neural Networks and $k$-Weisfeiler-Lehman tests

The Message Passing Neural Network (MPNN) is a generic model that incorporates many popular architectures, and it is based on learning local aggregations of information in the graph [17]. When applied to an undirected graph $G = (V, E, x, e)$, an MPNN with $T$ layers is defined iteratively as follows. For $t < T$, to compute the message $m_i^{(t+1)}$ and the hidden state $h_i^{(t+1)}$ for each node $i \in V$ at the $(t+1)$th layer, we apply the following update rule:

$$m_i^{(t+1)} = \sum_{\mathcal{N}(i)} M_t(h_i^{(t)}, h_j^{(t)}, e_{i,j}), \qquad h_i^{(t+1)} = U_t(h_i^{(t)}, m_i^{(t+1)}) \,,$$

where $\mathcal{N}(i)$ is the neighborhood of node $i$ in $G$, $M_t$ is the message function at layer $t$ and $U_t$ is the vertex update function at layer $t$. Finally, a graph-level prediction is computed as $\hat{y} = R(\{h_i^{(T)} : i \in V\})$, where $R$ is the readout function. Typically, the hidden states at the first layer are set as $h_i^{(0)} = x_i$. Learnable parameters can appear in the functions $M_t$, $U_t$ (for all $t \in [T]$) and $R$.

Xu et al. [64] and Morris et al. [44] show that, when the graphs' edges are unweighted, such models are at most as powerful as the Weisfeiler-Lehman (WL) test in distinguishing non-isomorphic graphs. Below, we will first prove an extension of this result that incorporates edge features. To do so, we first introduce the hierarchy of $k$-Weisfeiler-Lehman ($k$-WL) tests. The $k$-WL test takes a pair of graphs $G^{[1]}$ and $G^{[2]}$ and attempts to determine whether they are isomorphic. In a nutshell, for each of the graphs, the test assigns an initial color in some color space to every $k$-tuple in $V^k$ according to its *isomorphism type*, and then it updates the colorings iteratively by aggregating information among neighboring $k$-tuples. The test will terminate and return the judgement that the two graphs are not isomorphic if and only if at some iteration $t$, the coloring multisets differ. We refer the reader to Appendix C for a rigorous definition.

**Remark 3.1.** *For graphs with unweighted edges, 1-WL and 2-WL are known to have the same discriminative power [39]. For $k \geq 2$, it is known that $(k + 1)$-WL is strictly more powerful than $k$-WL, in the sense that there exist pairs of graph distinguishable by the former but not the latter [6]. Thus, with growing $k$, the set of $k$-WL tests forms a hierarchy with increasing discriminative power. Note that there has been an different definition of WL in the literature, sometimes known as Folklore Weisfeiler-Lehman (FWL), with different properties [39, 44].* [2]

Our first result is an extension of Morris et al. [44], Xu et al. [64] to incorporate edge features.

**Theorem 3.2.** *Two attributed graphs that are indistinguishable by 2-WL cannot be distinguished by any MPNN.*

The theorem is proven in Appendix D. Thus, it motivates us to first study what patterns 2-WL can or cannot count.

## 3.1 Substructure counting by 2-WL and MPNNs

It turns out that whether or not 2-WL can perform *induced-subgraph-count* of a pattern is completely characterized by the number of nodes in the pattern. Any connected pattern with 1 or 2 nodes (i.e., representing a node or an edge) can be easily counted by an MPNN with 0 and 1 layer of message-passing, respectively, or by 2-WL with 0 iterations[3]. In contrast, for all larger connected patterns, we have the following negative result, which we prove in Appendix E.

**Theorem 3.3.** *2-WL cannot induced-subgraph-count any connected pattern with 3 or more nodes.*

The intuition behind this result is that, given any connected pattern with 3 or more nodes, we can construct a pair of graphs that have different *induced-subgraph-counts* of the pattern but cannot be distinguished from each other by 2-WL, as illustrated in Figure 2. Thus, together with Theorem 3.2, we have

**Corollary 3.4.** *MPNNs cannot induced-subgraph-count any connected pattern with 3 or more nodes.*

For *subgraph-count*, if both nodes and edges are unweighted, Arvind et al. [2] show that the only patterns 1-WL (and equivalently 2-WL) can count are either star-shaped patterns and pairs of disjoint

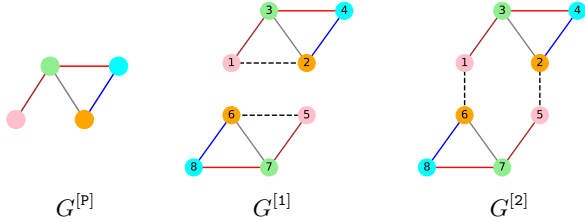

Figure 2: Illustration of the construction in the proof of Theorem 3.3 for the pattern $G^{[\text{P}]}$ on the left. Note that $\mathsf{C}_I(G^{[1]}; G^{[\text{P}]}) = 0$ and $\mathsf{C}_I(G^{[2]}; G^{[\text{P}]}) = 2$, and the graphs $G^{[1]}$ and $G^{[2]}$ cannot be distinguished by 2-WL, MPNNs or 2-IGNs.

$G^{[\text{P}]}$ $\qquad$ $G^{[1]}$ $\qquad$ $G^{[2]}$

edges. We prove the positive result that MPNNs can count star-shaped patterns even when node and edge features are allowed, utilizing a result in Xu et al. [64] that the message functions are able to approximate any function on multisets.

**Theorem 3.5.** *MPNNs can perform subgraph-count of star-shaped patterns.*

By Theorem 3.2, this implies that

**Corollary 3.6.** 2-*WL can perform subgraph-count of star-shaped patterns.*

### 3.2 Substructure counting by $k$-WL

There have been efforts to extend the power of GNNs by going after $k$-WL for higher $k$, such as Morris et al. [44]. Thus, it is also interesting to study the patterns that $k$-WL can and cannot count. Since $k$-tuples are assigned initial colors based on their isomorphism types, the following is easily seen, and we provide a proof in Appendix G.

**Theorem 3.7.** $k$-*WL, at initialization, is able to perform both induced-subgraph-count and subgraph-count of patterns consisting of at most $k$ nodes.*

This establishes a potential hierarchy of increasing power in terms of substructure counting by $k$-WL. However, tighter results can be much harder to achieve. For example, to show that 2-FWL (and therefore 3-WL) cannot count cycles of length 8, Fürer [15] has to rely on computers for counting cycles in the classical Cai-Fürer-Immerman counterexamples to $k$-WL [6]. We leave the pursuit of general and tighter characterizations of $k$-WL's substructure counting power for future research, but we are nevertheless able to prove a partial negative result concerning finite iterations of $k$-WL.

**Definition 3.8.** *A path pattern of size $m$, denoted by $H_m$, is an unattributed graph, $H_m = (V^{[\text{H}_\text{m}]}, E^{[\text{H}_\text{m}]})$, where $V^{[\text{H}_\text{m}]} = [m]$, and $E^{[\text{H}_\text{m}]} = \{(i, i+1) : 1 \le i < m\} \cup \{(i+1, i) : 1 \le i < m\}$.*

**Theorem 3.9.** *Running $T$ iterations of $k$-WL cannot perform induced-subgraph-count of any path pattern of $(k+1)2^T$ or more nodes.*

The proof is given in Appendix H. This bound grows quickly when $T$ becomes large. However, since in practice, many if not most GNN models are designed to be shallow [62, 74], this result is still relevant for studying finite-depth GNNs that are based on $k$-WL.

## 4 Invariant Graph Networks

Recently, diverging from the strategy of local aggregation of information as adopted by MPNNs and $k$-WLs, an alternative family of GNN models called *Invariant Graph Networks (IGNs)* was introduced in Maron et al. [39, 40, 41]. Here, we restate its definition. First, note that if the node and edge features are vectors of dimension $d_n$ and $d_e$, respectively, then an input graph can be represented by a *second-order tensor* $\boldsymbol{B} \in \mathbb{R}^{n \times n \times (d+1)}$, where $d = \max(d_n, d_e)$, defined by

$$\begin{aligned} \boldsymbol{B}_{i,i,1:d_n} &= x_i \,, \quad \forall i \in V = [n] \,, \\ \boldsymbol{B}_{i,j,1:d_e} &= e_{i,j} \,, \quad \forall (i,j) \in E \,, \\ \boldsymbol{B}_{1:n,1:n,d+1} &= A \,. \end{aligned} \tag{1}$$

If the nodes and edges do not have features, $\boldsymbol{B}$ simply reduces to the adjacency matrix. Thus, GNN models can be alternatively defined as functions on such second-order tensors. More generally, with graphs represented by $k$th-order tensors, we can define:

**Definition 4.1.** *A $k$th-order Invariant Graph Network ($k$-IGN) is a function $F : \mathbb{R}^{n^k \times d_0} \to \mathbb{R}$ that can be decomposed in the following way:*

$$F = m \circ h \circ L^{(T)} \circ \sigma \circ \cdots \circ \sigma \circ L^{(1)},$$

*where each $L^{(t)}$ is a linear equivariant layer [40] from $\mathbb{R}^{n^k \times d_{t-1}}$ to $\mathbb{R}^{n^k \times d_t}$, $\sigma$ is a pointwise activation function, $h$ is a linear invariant layer from $\mathbb{R}^{n^k \times d_T}$ to $\mathbb{R}$, and $m$ is an MLP.*

Maron et al. [41] show that if $k$ is allowed to grow as a function of the size of the graphs, then $k$-IGNs can achieve universal approximation of permutation-invariant functions on graphs. Nonetheless, due to the quick growth of computational complexity and implementation difficulty as $k$ increases, in practice it is hard to have $k > 2$. If $k = 2$, on one hand, it is known that 2-IGNs are at least as powerful as 2-WL [38]; on the other hand, 2-IGNs are not universal [8]. However, it remains open to establish a strict upper bound on the expressive power of 2-IGNs in terms of the WL tests as well as to characterize concretely their limitations. Here, we first answer the former question by proving that 2-IGNs are no more powerful than 2-WL:

**Lemma 4.2.** *If two graphs are indistinguishable by 2-WL, then no 2-IGN can distinguish them either.*

We give the full proof of Lemma 4.2 in Appendix I. As a consequence, we then have

**Corollary 4.3.** *2-IGNs are exactly as powerful as 2-WL.*

Thanks to this equivalence, the following results on the ability of 2-IGNs to count substructures are immediate corollaries of Theorem 3.3 and Corollary 3.6 (though we also provide a direct proof of Corollary 4.4 in Appendix J):

**Corollary 4.4.** *2-IGNs cannot perform induced-subgraph-count of any connected pattern with 3 or more nodes.*

**Corollary 4.5.** *2-IGNs can perform subgraph-count of star-shaped patterns.*

In addition, as $k$-IGNs are no less powerful than $k$-WL [39], as a corollary of Theorem 3.7, we have

**Corollary 4.6.** *$k$-IGNs can perform both induced-subgraph-count and subgraph-count of patterns consisting of at most $k$ nodes.*

## 5 Local Relational Pooling

While deep MPNNs and 2-IGNs are able to aggregate information from multi-hop neighborhoods, our results show that they are unable to preserve information such as the induced-subgraph-counts of nontrivial patterns. To bypass such limitations, we suggest going beyond the strategy of iteratively aggregating information in an equivariant way, which underlies both MPNNs and IGNs. One helpful observation is that, if a pattern is present in the graph, it can always be found in a sufficiently large local neighborhood, or *egonet*, of some node in the graph [50]. An egonet of depth $l$ centered at a node $i$ is the induced subgraph consisting of $i$ and all nodes within distance $l$ from it. Note that any pattern with radius $r$ is contained in some egonet of depth $l = r$. Hence, we can obtain a model capable of counting patterns by applying a powerful local model to each egonet separately and then aggregating the outputs across all egonets, as we will introduce below.

For such a local model, we adopt the Relational Pooling (RP) approach from Murphy et al. [45]. In summary, it creates a powerful permutation-invariant model by symmetrizing a powerful permutation-sensitive model, where the symmetrization is performed by averaging or summing over all permutations of the nodes' ordering. Formally, let $\boldsymbol{B} \in \mathbb{R}^{n \times n \times d}$ be a permutation-sensitive second-order tensor representation of the graph $G$, such as the one defined in (1). Then, an RP model is defined by

$$f_{\text{RP}}(G) = \frac{1}{|S_n|} \sum_{\pi \in S_n} f(\pi \star \boldsymbol{B}),$$

where $f$ is some function that is not necessarily permutation-invariant, such as a general multi-layer perceptron (MLP) applied to the vectorization of its tensorial input, $S_n$ is the set of permutations on $n$ nodes, and $\pi \star \boldsymbol{B}$ is $\boldsymbol{B}$ transformed by permuting its first two dimensions according to $\pi$, i.e., $(\pi \star \boldsymbol{B})_{j_1,j_2,p} = \boldsymbol{B}_{\pi(j_1),\pi(j_2),p}$. For choices of $f$ that are sufficiently expressive, such $f_{\text{RP}}$'s are shown to be an universal approximator of permutation-invariant functions [45]. However, the summation quickly becomes intractable once $n$ is large, and hence approximation methods have been introduced. In comparison, since we apply this model to small egonets, it is tractable to compute the model exactly. Moreover, as egonets are rooted graphs, we can reduce the symmetrization over all permutations in $S_n$ to the subset $S_n^{\text{BFS}} \subseteq S_n$ of permutations which order the nodes in a way that

is compatible with breath-first-search (BFS), as suggested in Murphy et al. [45] to further reduce the complexity. Defining $G_{i,l}^{[\text{ego}]}$ as the egonet centered at node $i$ of depth $l$, $\boldsymbol{B}_{i,l}^{[\text{ego}]}$ as the tensor representation of $G_{i,l}^{[\text{ego}]}$ and $n_{i,l}$ as the number of nodes in $G_{i,l}^{[\text{ego}]}$, we consider models of the form

$$f_{\text{LRP}}^l(G) = \sum_{i \in V} H_i, \qquad H_i = \frac{1}{|S_{n_{i,l}}^{\text{BFS}}|} \sum_{\pi \in S_{n_{i,l}}^{\text{BFS}}} f\left(\pi \star \boldsymbol{B}_{i,l}^{[\text{ego}]}\right), \tag{2}$$

To further improve efficiency, we propose to only consider *ordered subsets* of the nodes in each egonet that are compatible with *$k$-truncated-BFS* rather than all orderings of the full node set of the egonet, where we define $k$-truncated-BFS to be a BFS-like procedure that only adds at most $k$ children of every node to the priority queue for future visits and uses zero padding when fewer than $k$ children have not been visited. We let $\tilde{S}_{i,l}^{k\text{-BFS}}$ denote the set of order subsets of the nodes in $G_{i,l}^{[\text{ego}]}$ that are compatible with $k$-truncated-BFS. Each $\tilde{\pi} \in \tilde{S}_{i,l}^{k\text{-BFS}}$ can be written as the ordered list $[\tilde{\pi}(1), ..., \tilde{\pi}(|\tilde{\pi}|)]$, where $|\tilde{\pi}|$ is the length of $\tilde{\pi}$, and for $i \in [|\tilde{\pi}|]$, each $\tilde{\pi}(i)$ is the index of a distinct node in $G_{i,l}^{[\text{ego}]}$. In addition, for each $\tilde{\pi} \in \tilde{S}_{i,l}^{k\text{-BFS}}$, we introduce a learnable normalization factor, $\alpha_{\tilde{\pi}}$, which can depend on the degrees of the nodes that appear in $\tilde{\pi}$, to adjust for the effect that adding irrelevant edges can alter the fraction of permutations in which a substructure of interest appears. It is a vector whose dimension matches the output dimension of $f$. More detail on this factor will be given below. Using $\odot$ to denote the element-wise product between vectors, our model becomes

$$f_{\text{LRP}}^{l,k}(G) = \sum_{i \in V} H_i, \qquad H_i = \frac{1}{|\tilde{S}_{i,l}^{k\text{-BFS}}|} \sum_{\tilde{\pi} \in \tilde{S}_{i,l}^{k\text{-BFS}}} \alpha_{\tilde{\pi}} \odot f\left(\tilde{\pi} \star \boldsymbol{B}_{i,l}^{[\text{ego}]}\right). \tag{3}$$

We call this model depth-$l$ size-$k$ *Local Relational Pooling (LRP-$l$-$k$)*. Depending on the task, the summation over all nodes can be replaced by taking average in the definition of $f_{\text{LRP}}^l(G)$. In this work, we choose $f$ to be an MLP applied to the vectorization of its tensorial input. For fixed $l$ and $k$, if the node degrees are upper-bounded, the time complexity of the model grows linearly in $n$.

In the experiments below, we focus on two particular variants, where either $l = 1$ or $k = 1$. When $l = 1$, we let $\alpha_{\tilde{\pi}}$ be the output of an MLP applied to the degree of root node $i$. When $k = 1$, note that each $\tilde{\pi} \in \tilde{S}_{i,l}^{k\text{-BFS}}$ consists of nodes on a path of length at most $(l + 1)$ starting from node $i$, and we let $\alpha_{\tilde{\pi}}$ be the output of an MLP applied to the concatenation of the degrees of all nodes on the path. More details on the implementations are discussed in Appendix M.1.

Furthermore, the LRP procedure can be applied iteratively in order to utilize multi-scale information. We define a *Deep LRP-$l$-$k$* model of $T$ layers as follows. For $t \in [T]$, we iteratively compute

$$H_i^{(t)} = \frac{1}{|\tilde{S}_{i,l}^{k\text{-BFS}}|} \sum_{\tilde{\pi} \in \tilde{S}_{i,l}^{k\text{-BFS}}} \alpha_{\tilde{\pi}}^{(t)} \odot f^{(t)}\left(\tilde{\pi} \star \boldsymbol{B}_{i,l}^{[\text{ego}]}(H^{(t-1)})\right), \tag{4}$$

where for an $H \in \mathbb{R}^{n \times d}$, $\boldsymbol{B}_{i,l}^{[\text{ego}]}(H)$ is the subtensor of $\boldsymbol{B}(H) \in \mathbb{R}^{n \times n \times d}$ corresponding to the subset of nodes in the egonet $G_{i,l}^{[\text{ego}]}$, and $\boldsymbol{B}(H)$ is defined by replacing each $x_i$ by $H_i$ in (1). The dimensions of $\alpha_{\tilde{\pi}}^{(t)}$ and the output of $f^{(t)}$ are both $d^{(t)}$, and we set $H_i^{(0)} = x_i$. Finally, we define the graph-level output to be, depending on the task,

$$f_{\text{DLRP}}^{l,k,T}(G) = \sum_{i \in V} H_i^{(T)} \qquad \text{or} \qquad \frac{1}{|V|} \sum_{i \in V} H_i^{(T)}. \tag{5}$$

The efficiency in practice can be greatly improved by leveraging a pre-computation of the set of maps $H \mapsto \tilde{\pi} \star \boldsymbol{B}_{i,l}^{[\text{ego}]}(H)$ as well as sparse tensor operations, which we describe in Appendix K.

## 6 Experiments

### 6.1 Counting substructures in random graphs

We first complement our theoretical results with numerical experiments on counting the five substructures illustrated in Figure 3 in synthetic random graphs, including the *subgraph-count* of 3-stars and

the *induced-subgraph-counts* of triangles, tailed triangles, chordal cycles and attributed triangles. By Theorem 3.3 and Corollary 3.4, MPNNs and 2-IGNs cannot exactly solve the *induced-subgraph-count* tasks; while by Theorem 3.5 and Corollary 4.5, they are able to express the *subgraph-count* of 3-stars.

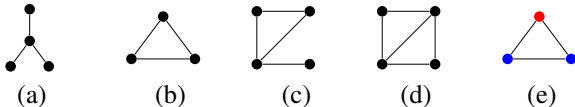

Figure 3: Substructures considered in the experiments: (a) 3-star (b) triangle (c) tailed triangle (d) chordal cycle (e) attributed triangle.

**Datasets.** We create two datasets of random graphs, one consisting of Erdős-Renyi graphs and the other random regular graphs. Further details on the generation of these datasets are described in Appendix M.2.1. As the target labels, we compute the ground-truth counts of these unattributed and attributed patterns in each graph with a counting algorithm proposed by Shervashidze et al. [56].

**Models**. We consider LRP, GraphSAGE (using full 1-hop neighborhood) [18], GIN [64], GCN [31], 2-IGN [40], PPGN [39] and spectral GNN (sGNN) [7], with GIN and GCN under the category of MPNNs. Details of the model architectures are given in Appendix M.1. We use mean squared error (MSE) for regression loss. Each model is trained on 1080ti five times with different random seeds.

**Results**. The results on the *subgraph-count* of 3-stars and the *induced-subgraph-count* of triangles are shown in Table 1, measured by the MSE on the test set divided by the variance of the ground truth counts of the pattern computed over all graphs in the dataset, while the results on the other counting tasks are shown in Appendix M.2.2. Firstly, the almost-negligible errors of LRP on all the tasks support our theory that LRP exploiting only egonets of depth 1 is powerful enough for counting patterns with radius 1. Moreover, GIN, 2-IGN and sGNN yield small error for the 3-star task compared to the variance of the ground truth counts, which is consistent with their theoretical power to perform subgraph-count of star-shaped patterns. Relative to the variance of the ground truth counts, GraphSAGE, GIN and 2-IGN have worse top performance on the triangle task than on the 3-star task, which is also expected from the theory (see Appendix L for a discussion on GraphSAGE). PPGN [39] with provable 3-WL discrimination power also performs well on both counting triangles and 3-stars. Moreover, the results provide interesting insights into the average-case performance in the substructure counting tasks, which are beyond what our theory can predict.

Table 1: Performance of different GNNs on learning the induced-subgraph-count of triangles and the subgraph-count of 3-stars on the two datasets, measured by test MSE divided by variance of the ground truth counts. Shown here are the best and the median (i.e., third-best) performances of each model over five runs with different random seeds. Values below 1E-3 are emboldened. Note that we select the best out of four variants for each of GCN, GIN, sGNN and GraphSAGE, and the better out of two variants for 2-IGN. Details of the GNN architectures and the results on the other counting tasks can be found in Appendices M.1 and M.2.2.

| | Erdős-Renyi | | | | Random Regular | | | |
|---|---|---|---|---|---|---|---|---|
| | triangle | | 3-star | | triangle | | 3-star | |
| | best | median | best | median | best | median | best | median |
| GCN | 6.78E-1 | 8.27E-1 | 4.36E-1 | 4.55E-1 | 1.82 | 2.05 | 2.63 | 2.80 |
| GIN | 1.23E-1 | 1.25E-1 | **1.62E-4** | **3.44E-4** | 4.70E-1 | 4.74E-1 | **3.73E-4** | **4.65E-4** |
| GraphSAGE | 1.31E-1 | 1.48E-1 | **2.40E-10** | **1.96E-5** | 3.62E-1 | 5.21E-1 | **8.70E-8** | **4.61E-6** |
| sGNN | 9.25E-2 | 1.13E-1 | 2.36E-3 | 7.73E-3 | 3.92E-1 | 4.43E-1 | 2.37E-2 | 1.41E-1 |
| 2-IGN | 9.83E-2 | 9.85E-1 | **5.40E-4** | 5.12E-2 | 2.62E-1 | 5.96E-1 | 1.19E-2 | 3.28E-1 |
| PPGN | **5.08E-8** | **2.51E-7** | **4.00E-5** | **6.01E-5** | **1.40E-6** | **3.71E-5** | **8.49E-5** | **9.50E-5** |
| LRP-1-3 | **1.56E-4** | **2.49E-4** | **2.17E-5** | **5.23E-5** | **2.47E-4** | **3.83E-4** | **1.88E-6** | **2.81E-6** |
| Deep LRP-1-3 | **2.81E-5** | **4.77E-5** | **1.12E-5** | **3.78E-5** | **1.30E-6** | **5.16E-6** | **2.07E-6** | **4.97E-6** |

## 6.2 Molecular prediction tasks

We evaluate LRP on the molecular prediction datasets ogbg-molhiv [63], QM9 [54] and ZINC [14]. More details of the setup can be found in Appendix M.3.

**Results**. The results on the ogbg-molhiv, QM9 and ZINC are shown in Tables 2 - 4, where for each task or target, the top performance is colored red and the second best colored violet. On ogbg-molhiv, Deep LRP-1-3 with early stopping (see Appendix M.3 for details) achieves higher testing ROC-AUC than the baseline models. On QM9, Deep LRP-1-3 and Deep-LRP-5-1 consistently outperform MPNN and achieve comparable performances with baseline models that are more powerful than

1-WL, including 123-gnn and PPGN. In particular, Deep LRP-5-1 attains the lowest test error on three targets. On ZINC, Deep LRP-7-1 achieves the best performance among the models that do not use feature augmentation (the top baseline model, GateGCN-E-PE, additionally augments the node features with the top Laplacian eigenvectors).

Table 2: Performances on ogbg-molhiv measured by ROC-AUC (%). [†]: Reported on the OGB learderboard [20]. [‡]: Reported in [21].

| Model | Training | Validation | Testing |
|---|---|---|---|
| GIN[†] | 88.64±2.54 | 82.32±0.90 | 75.58±1.40 |
| GIN + VN[†] | 92.73±3.80 | 84.79±0.68 | **77.07±1.49** |
| GCN[†] | 88.54±2.19 | 82.04±1.41 | 76.06±0.97 |
| GCN + VN[†] | 90.07±4.69 | 83.84±0.91 | 75.99±1.19 |
| GAT [59][‡] | - | - | 72.9±1.8 |
| GraphSAGE [18][‡] | - | - | 74.4±0.7 |
| Deep LRP-1-3 | 89.81±2.90 | 81.31±0.88 | 76.87±1.80 |
| Deep LRP-1-3 (ES) | 87.56±2.11 | 82.09±1.16 | **77.19±1.40** |

Table 3: Performances on ZINC measured by the Mean Absolute Error (MAE). †: Reported in Dwivedi et al. (2020).

| Model | Training | Testing | Time / Ep |
|---|---|---|---|
| GraphSAGE[†] | $0.081 \pm 0.009$ | $0.398 \pm 0.002$ | 16.61s |
| GIN[†] | $0.319 \pm 0.015$ | $0.387 \pm 0.015$ | 2.29s |
| MoNet[†] | $0.093 \pm 0.014$ | $0.292 \pm 0.006$ | 10.82s |
| GatedGCN-E[†] | $0.074 \pm 0.016$ | $0.282 \pm 0.015$ | 20.50s |
| GatedGCN-E-PE[†] | $0.067 \pm 0.019$ | **$0.214 \pm 0.013$** | 10.70s |
| PPGN[†] | $0.140 \pm 0.044$ | $0.256 \pm 0.054$ | 334.69s |
| Deep LRP-7-1 | $0.028 \pm 0.004$ | **$0.223 \pm 0.008$** | 72s |
| Deep LRP-5-1 | $0.020 \pm 0.006$ | $0.256 \pm 0.033$ | 42s |

Table 4: Performances on QM9 measured by the testing Mean Absolute Error. All baseline results are from [39], including DTNN [63] and 123-gnn [44]. The loss value on the last row is defined in Appendix M.3.

| Target | DTNN | MPNN | 123-gnn | PPGN | Deep LRP-1-3 | Deep LRP-5-1 |
|---|---|---|---|---|---|---|
| $\mu$ | **0.244** | 0.358 | 0.476 | **0.231** | 0.399 | **0.298** |
| $\alpha$ | 0.95 | 0.89 | **0.27** | 0.382 | 0.337 | **0.298** |
| $\epsilon_{homo}$ | 0.00388 | 0.00541 | 0.00337 | **0.00276** | 0.00287 | **0.00254** |
| $\epsilon_{lumo}$ | 0.00512 | 0.00623 | 0.00351 | **0.00287** | 0.00309 | **0.00277** |
| $\Delta_\epsilon$ | 0.0112 | 0.0066 | 0.0048 | 0.00406 | **0.00396** | **0.00353** |
| $\langle R^2 \rangle$ | **17** | 28.5 | 22.9 | **16.07** | 20.4 | 19.3 |
| ZPVE | 0.00172 | 0.00216 | **0.00019** | 0.00064 | 0.00067 | **0.00055** |
| $U_0$ | 2.43 | 2.05 | **0.0427** | 0.234 | 0.590 | 0.413 |
| U | 2.43 | 2 | **0.111** | 0.234 | 0.588 | 0.413 |
| H | 2.43 | 2.02 | **0.0419** | 0.229 | 0.587 | 0.413 |
| G | 2.43 | 2.02 | **0.0469** | 0.238 | 0.591 | 0.413 |
| $C_v$ | 0.27 | 0.42 | **0.0944** | 0.184 | 0.149 | **0.129** |
| Loss | 0.1014 | 0.1108 | 0.0657 | **0.0512** | 0.0641 | **0.0567** |

## 7 Conclusions

We propose a theoretical framework to study the expressive power of classes of GNNs based on their ability to count substructures. We distinguish two kinds of counting: subgraph-count and induced-subgraph-count. We prove that neither MPNNs nor 2-IGNs can induced-subgraph-count any connected structure with 3 or more nodes; $k$-IGNs and $k$-WL can subgraph-count and induced-subgraph-count any pattern of size $k$. We also provide an upper bound on the size of "path-shaped" substructures that finite iterations of $k$-WL can induced-subgraph-count. To establish these results, we prove an equivalence between approximating graph functions and discriminating graphs. Also, as intermediary results, we prove that MPNNs are no more powerful than 2-WL on attributed graphs, and that 2-IGNs are equivalent to 2-WL in distinguishing non-isomorphic graphs, which partly answers an open problem raised in Maron et al. [38]. In addition, we perform numerical experiments that support our theoretical results and show that the Local Relational Pooling approach inspired by Murphy et al. [45] can successfully count certain substructures. In summary, we build the foundation for using substructure counting as an intuitive and relevant measure of the expressive power of GNNs, and our concrete results for existing GNNs motivate the search for more powerful designs of GNNs.

One limitation of our theory is that it is only concerned with the expressive power of GNNs and no their optimization or generalization. Our theoretical results are also worse-case in nature and cannot predict average-case performance. Many interesting theoretical questions remain, including better characterizing the ability to count substructures of general $k$-WL and $k$-IGNs as well as other architectures such as spectral GNNs [7] and polynomial IGNs [38]. On the practical side, we hope our framework can help guide the search for more powerful GNNs by considering substructure counting as a criterion. It will be interesting to quantify the relevance of substructure counting in empirical tasks, perhaps following the work of Ying et al. [68], and also to consider tasks where substructure counting is explicitly relevant, such as subgraph matching [36].

## Broader impact

In this work we propose to understand the power of GNN architectures via the substructures that they can and cannot count. Our work is motivated by the relevance of detecting and counting *graph substructures* in applications, and the current trend on using deep learning – in particular, graph neural networks – in such scientific fields. The ability of different GNN architectures to count graph substructures not only serves as an intuitive theoretical measure of their expressive power but also is highly relevant to real-world scenarios. Our results show that some widely used GNN architectures are not able to count substructures. Such knowledge may indicate that some widely-used graph neural network architectures are actually not the right tool for certain scientific problems. On the other hand, we propose a GNN model that not only has the ability to count substructures but also can learn from data what the relevant substructures are.

## Acknowledgements

We are grateful to Haggai Maron, Jiaxuan You, Ryoma Sato and Christopher Morris for helpful conversations. This work is partially supported by the Alfred P. Sloan Foundation, NSF RI-1816753, NSF CAREER CIF 1845360, NSF CHS-1901091, Samsung Electronics, and the Institute for Advanced Study. SV is supported by NSF DMS 2044349, EOARD FA9550-18-1-7007, and the NSF-Simons Research Collaboration on the Mathematical and Scientific Foundations of Deep Learning (MoDL) (NSF DMS 2031985).

## Footnotes

[1]Code available at `https://github.com/leichen2018/GNN-Substructure-Counting`.

[2]When "WL test" is used in the literature without specifying "$k$", it usually refers to 1-WL, 2-WL or 1-FWL.

[3]In fact, this result is a special case of Theorem 3.7.

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
