[Supplementary Material]

# A  Function approximation perspective of substructure counting

On a space of graphs $\mathcal{G}$, we call $\mathsf{C}_I(\cdot; G^{[\mathrm{P}]})$ the *induced-subgraph-count function* of the pattern $G^{[\mathrm{P}]}$, and $\mathsf{C}_S(\cdot; G^{[\mathrm{P}]})$ the *subgraph-count function* of $G^{[\mathrm{P}]}$. To formalize the probe into whether certain GNN architectures can count different substructures, a natural question to study is whether they are able to approximate the induced-subgraph-count and the subgraph-count functions arbitrarily well. Formally, given a target function $g : \mathcal{G} \to \mathbb{R}$, and family of functions, $\mathcal{F}$, which in our case is typically the family of functions that a GNN architecture can represent, we say $\mathcal{F}$ is able to approximate $g$ on $\mathcal{G}$ if for all $\epsilon > 0$ there exists $f \in \mathcal{F}$ such that $|g(G) - f(G)| < \epsilon$, for all $G \in \mathcal{G}$.

However, such criterion based on function approximation is hard to work with directly when we look at concrete examples later on. For this reason, below we will look for an alternative and equivalent definition from the perspective of graph discrimination.

## A.1  From function approximation to graph discrimination

Say $\mathcal{G}$ is a space of graphs, and $\mathcal{F}$ is a family of functions from $\mathcal{G}$ to $\mathbb{R}$. Given two graphs $G^{[1]}, G^{[2]} \in \mathcal{G}$, we say $\mathcal{F}$ is able to distinguish them if there exists $f \in \mathcal{F}$ such that $f(G^{[1]}) \neq f(G^{[2]})$. Such a perspective has been explored in Chen et al. [8], for instance, to build an equivalence between function approximation and graph isomorphism testing by GNNs. In the context of substructure counting, it is clear that the ability to approximate the count functions entails the ability to distinguish graphs in the following sense:

**Observation 1.** *If $\mathcal{F}$ is able to approximate the induced-subgraph-count (or subgraph-count) function of a pattern $G^{[\mathrm{P}]}$ on the space $\mathcal{G}$, then for all $G^{[1]}, G^{[2]} \in \mathcal{G}$ such that $\mathsf{C}_I(G^{[1]}, G^{[\mathrm{P}]}) \neq \mathsf{C}_I(G^{[2]}, G^{[\mathrm{P}]})$ (or $\mathsf{C}_S(G^{[1]}, G^{[\mathrm{P}]}) \neq \mathsf{C}_S(G^{[2]}, G^{[\mathrm{P}]})$), they can be distinguished by $\mathcal{F}$.*

What about the converse? When the space $\mathcal{G}$ is finite, such as if the graphs have bounded numbers of nodes and the node as well as edge features belong to finite alphabets, we can show a slightly weaker statement than the exact converse. Following Chen et al. [8], we define an augmentation of families of functions using feed-forward neural networks as follows:

**Definition A.1.** *Given $\mathcal{F}$, a family of functions from a space $\mathcal{X}$ to $\mathbb{R}$, we consider an augmented family of functions also from $\mathcal{X}$ to $\mathbb{R}$ consisting of all functions of the following form*

$$x \mapsto h_{\mathcal{NN}}([f_1(x), ..., f_d(x)]),$$

*where $d \in \mathbb{N}$, $h_1, ..., h_d \in \mathcal{F}$, and $h_{\mathcal{NN}}$ is a feed-forward neural network / multi-layer perceptron. When $\mathcal{NN}$ is restricted to have $L$ layers at most, we denote this augmented family by $\mathcal{F}^{+L}$.*

**Lemma A.2.** *Suppose $\mathcal{X}$ is a finite space, $g$ is a finite function on $\mathcal{X}$, and $\mathcal{F}$ is a family of functions on $\mathcal{X}$. Then, $\mathcal{F}^{+1}$ is able to approximate $f$ on $\mathcal{G}$ if $\forall x_1, x_2 \in \mathcal{X}$ with $g(x_1) \neq g(x_2)$, $\exists f \in \mathcal{F}$ such that $f(x_1) \neq f(x_2)$.*

*Proof.* Since $\mathcal{X}$ is a finite space, for some large enough integer $d$, $\exists$ a collection of $d$ functions, $f_1, ..., f_d \in \mathcal{F}$ such that, if we define the function $\mathbf{f}(x) = (f_1(x), ..., f_d(x)) \in \mathbb{R}^d$, then it holds that $\forall x_1, x_2 \in \mathcal{X}, \mathbf{f}(x_1) = \mathbf{f}(x_2) \Rightarrow g(x_1) = g(x_2)$. (In fact, we can choose $d \leq \frac{|\mathcal{X}| \cdot (|\mathcal{X}| - 1)}{2}$, since in the worst case we need one $f_i$ per pair of $x_1, x_2 \in \mathcal{X}$ with $x_1 \neq x_2$.) Then, $\exists$ a well-defined function $h$ from $\mathbb{R}^d$ to $\mathbb{R}$ such that $\forall x \in \mathcal{X}, g(x) = h(\mathbf{f}(x))$. By the universal approximation power of neural networks, $h$ can then be approximated arbitrarily well by some neural network $h_{\mathcal{NN}}$. $\qquad \square$

Thus, in the context of substructure counting, we have the following observation.

**Observation 2.** *Suppose $\mathcal{G}$ is a finite space. If $\forall G^{[1]}, G^{[2]} \in \mathcal{G}$ with $\mathsf{C}_I(G^{[1]}, G^{[\mathrm{P}]}) \neq \mathsf{C}_I(G^{[2]}, G^{[\mathrm{P}]})$ (or $\mathsf{C}_S(G^{[1]}, G^{[\mathrm{P}]}) \neq \mathsf{C}_S(G^{[2]}, G^{[\mathrm{P}]})$), $\mathcal{F}$ is able to distinguish $G^{[1]}$ and $G^{[2]}$, then $\mathcal{F}^{+1}$ is able to approximate the induced-subgraph-count (or subgraph-count) function of the pattern $G^{[\mathrm{P}]}$ on $\mathcal{G}$.*

For many GNN families, $\mathcal{F}^{+1}$ in fact has the same expressive power as $\mathcal{F}$. For example, consider $\mathcal{F}_{\mathrm{MPNN}}$, the family of all Message Passing Neural Networks on $\mathcal{G}$. $\mathcal{F}_{\mathrm{MPNN}}^{+1}$ consists of functions that run several MPNNs on the input graph in parallel and stack their outputs to pass through an MLP. However, running several MPNNs in parallel is equivalent to running one MPNN with larger dimensions of hidden states and messages, and moreover the additional MLP at the end can be merged into the readout function. Similar holds for the family of all $k$-Invariant Graph Functions ($k$-IGNs). Hence, for such GNN families, we have an exact equivalence on finite graph spaces $\mathcal{G}$.

## B    Additional notations

For two positive integers $a$ and $b$, we define $\text{MOD}_a(b)$ to be $a$ if $a$ divides $b$ and the number $c$ such that $b \equiv c \pmod{a}$ otherwise. Hence the value ranges from 1 to $a$ as we vary $b \in \mathbb{N}^*$.

For a positive integer $c$, let $[c]$ denote the set $\{1, ..., c\}$.

Two $k$-typles, $(i_i, ..., i_k), (j_1, ..., j_k) \in V^k$ are said to be in the same *equivalent class* if $\exists$ a permutation $\pi$ on $V$ such that $(\pi(i_i), ..., \pi(i_k)) = (j_1, ..., j_k)$. Note that belonging to the same equivalence class is a weaker condition than having the same isomorphism type, as will be defined in Appendix C, which has to do with what the graphs look like.

For any $k$-tuple, $s = (i_1, ..., i_k)$, and for $w \in [k]$, use $\texttt{I}_w(s)$ to denote the $w$th entry of $s$, $i_w$.

## C    Definition of $k$-WL on attributed graphs

In this section, we introduce the general $k$-WL test for $k \in \mathbb{N}^*$ applied to a pair of graphs, $G^{[1]}$ and $G^{[2]}$. Assume that the two graphs have the same number of vertices, since otherwise they can be told apart easily. Without loss of generality, we assume that they share the same set of vertex indices, $V$ (but can differ in $E$, $x$ or $e$). For each of the graphs, at iteration 0, the test assigns an initial color in some color space to every $k$-tuple in $V^k$ according to its isomorphism type (we define isomorphism types rigorously in Section C.1), and then updates the coloring in every iteration. For any $k$-tuple $s = (i_1, ..., i_k) \in V^k$, we let $c_k^{(t)}(s)$ denote the color of $s$ in $G^{[1]}$ assigned at $t$th iteration, and let $c'^{(t)}_k(s)$ denote the color it receives in $G^{[2]}$. $c_k^{(t)}(s)$ and $c'^{(t)}_k(s)$ are updated iteratively as follows. For each $w \in [k]$, define the neighborhood

$$N_w(s) = \{(i_1, ..., i_{w-1}, j, i_{j+1}, ..., i_k) : j \in V\}$$

Given $c_k^{(t-1)}$ and $c'^{(t-1)}_k$, define

$$C_w^{(t)}(s) = \text{HASH}_{t,1}\left(\{c_k^{(t-1)}(\tilde{s}) : \tilde{s} \in N_w(s)\}\right)$$

$$C'^{(t)}_w(s) = \text{HASH}_{t,1}\left(\{c'^{(t-1)}_k(\tilde{s}) : \tilde{s} \in N_w(s)\}\right)$$

with "$\{\}$" representing a multiset, and $\text{HASH}_{t,1}$ being some hash function that maps injectively from the space of multisets of colors to some intermediate space. Then let

$$c_k^{(t)}(s) = \text{HASH}_{t,2}\left(\left(c_k^{(t-1)}(s), \left(C_1^{(t)}(s), ..., C_k^{(t)}(s)\right)\right)\right)$$

$$c'^{(t)}_k(s) = \text{HASH}_{t,2}\left(\left(c'^{(t-1)}_k(s), \left(C'^{(t)}_1(s), ..., C'^{(t)}_k(s)\right)\right)\right)$$

where $\text{HASH}_{t,2}$ maps injectively from its input space to the space of colors. The test will terminate and return the result that the two graphs are not isomorphic if at some iteration $t$, the following two multisets differ:

$$\{c_k^{(t)}(s) : s \in V^k\} \neq \{c'^{(t)}_k(s) : s \in V^k\}$$

### C.1    Isomorphism types of $k$-tuples in $k$-WL for attributed graphs

Say $G^{[1]} = (V^{[1]}, E^{[1]}, x^{[1]}, e^{[1]})$, $G^{[2]} = (V^{[2]}, E^{[2]}, x^{[2]}, e^{[2]})$.

a) $\forall s = (i_1, ..., i_k), s' = (i'_1, ..., i'_k) \in (V^{[1]})^k$, $s$ and $s'$ are said to have the same isomorphism type if

    1. $\forall \alpha, \beta \in [k], i_\alpha = i_\beta \Leftrightarrow i'_\alpha = i'_\beta$

    2. $\forall \alpha \in [k], x^{[1]}_{i_\alpha} = x^{[1]}_{i'_\alpha}$

3. $\forall \alpha, \beta \in [k], (i_\alpha, i_\beta) \in E^{[1]} \Leftrightarrow (i'_\alpha, i'_\beta) \in E^{[1]}$, and moreover, if either side is true, then $e^{[1]}_{i_\alpha, i_\beta} = e^{[1]}_{i'_\alpha, i'_\beta}$

b) Similar if both $s, s' \in (V^{[2]})^k$.

c) $\forall s = (i_1, ..., i_k) \in (V^{[1]})^k, s' = (i'_1, ..., i'_k) \in (V^{[2]})^k$, $s$ and $s'$ are said to have the same isomorphism type if

1. $\forall \alpha, \beta \in [k], i_\alpha = i_\beta \Leftrightarrow i'_\alpha = i'_\beta$

2. $\forall \alpha \in [k], x^{[1]}_{i_\alpha} = x^{[2]}_{i'_\alpha}$

3. $\forall \alpha, \beta \in [k], (i_\alpha, i_\beta) \in E^{[1]} \Leftrightarrow (i'_\alpha, i'_\beta) \in E^{[2]}$, and moreover, if either side is true, then $e^{[1]}_{i_\alpha, i_\beta} = e^{[2]}_{i'_\alpha, i'_\beta}$

In $k$-WL tests, two $k$-tuples $s$ and $s'$ in either $(V^{[1]})^k$ or $(V^{[2]})^k$ are assigned the same color at iteration 0 if and only if they have the same isomorphism type.

For a reference, see Maron et al. [39].

## D   Proof of Theorem 3.2 (MPNNs are no more powerful than 2-WL)

*Proof.* Suppose for contradiction that there exists an MPNN with $T_0$ layers that can distinguish the two graphs. Let $m^{(t)}$ and $h^{(t)}$, $m'^{(t)}$ and $h'^{(t)}$ be the messages and hidden states at layer $t$ obtained by applying the MPNN on the two graphs, respectively. Define

$$\tilde{h}^{(t)}_{i,j} = \begin{cases} h^{(t)}_i & \text{if } i = j \\ \left( h^{(t)}_i, h^{(t)}_j, a_{i,j}, e_{i,j} \right) & \text{otherwise} \end{cases}$$

$$\tilde{h}'^{(t)}_{i,j} = \begin{cases} h'^{(t)}_i & \text{if } i = j \\ \left( h'^{(t)}_i, h'^{(t)}_j, a'_{i,j}, e'_{i,j} \right) & \text{otherwise,} \end{cases}$$

where $a_{i,j} = 1$ if $(i, j) \in E^{[1]}$ and 0 otherwise, $e_{i,j} = e^{[1]}_{i,j}$ is the edge feature of the first graph, and $a', e'$ are defined similarly for the second graph.

Since the two graphs cannot be distinguished by 2-WL, then for the $T_0$th iteration, there is

$$\{c^{(T_0)}_2(s) : s \in V^2\} = \{c'^{(T_0)}_2(s) : s \in V^2\},$$

which implies that there exists a permutation on $V^2$, which we can call $\eta_0$, such that $\forall s \in V^2$, there is $c^{(T_0)}_2(s) = c'^{(T_0)}_2(\eta_0(s))$. To take advantage of this condition, we introduce the following lemma, which is central to the proof.

**Lemma D.1.** $\forall t \leq T_0, \forall i, j, i', j' \in V$, if $c^{(t)}_2((i, j)) = c'^{(t)}_2((i', j'))$, then

1. $i = j \Leftrightarrow i' = j'$.

2. $\tilde{h}^{(t)}_{i,j} = \tilde{h}'^{(t)}_{i',j'}$

*Proof of Lemma D.1:* First, we state the following simple observation without proof, which is immediate given the update rule of $k$-WL:

**Lemma D.2.** For $k$-WL, $\forall s, s' \in V^k$, if for some $t_0$, $c^{(t_0)}_k(s) = c'^{(t_0)}_k(s')$, then $\forall t \in [0, t_0]$, $c^{(t)}_k(s) = c'^{(t)}_k(s')$.

For the first condition, assuming $\boldsymbol{c}_2^{(t)}((i,j)) = \boldsymbol{c'}_2^{(t)}((i',j'))$, Lemma D.2 then tells us that $\boldsymbol{c}_2^{(0)}((i,j)) = \boldsymbol{c'}_2^{(0)}((i',j'))$. Since the colors in 2-WL are initialized by the isomorphism type of the node pair, it has to be that $i = j \Leftrightarrow i' = j'$.

We will prove the second condition by induction on $t$. For the base case, $t = 0$, we want to show that $\forall i, j, i', j' \in V$, if $\boldsymbol{c}_2^{(0)}((i,j)) = \boldsymbol{c'}_2^{(0)}((i',j'))$ then $\tilde{h}_{i,j}^{(0)} = \tilde{h'}_{i',j'}^{(0)}$. If $i = j$, then $\boldsymbol{c}_2^{(0)}((i,i)) = \boldsymbol{c'}_2^{(0)}((i',i'))$ if and only if $x_i = x'_{i'}$, which is equivalent to $h_i^{(0)} = h'^{(0)}_{i'}$, and hence $\tilde{h}_i^{(0)} = \tilde{h'}_{i'}^{(0)}$. If $i \neq j$, then by the definition of isomorphism types given in Appendix C, $\boldsymbol{c}_2^{(0)}((i,j)) = \boldsymbol{c'}_2^{(0)}((i',j'))$ implies that

$$x_i = x'_{i'} \Rightarrow h_i^{(0)} = h'^{(0)}_{i'}$$
$$x_j = x'_{j'} \Rightarrow h_j^{(0)} = h'^{(0)}_{j'}$$
$$a_{i,j} = a'_{i',j'}$$
$$e_{i,j} = e'_{i',j'}$$

which yields $\tilde{h}_{i,j}^{(0)} = \tilde{h'}_{i',j'}^{(0)}$.

Next, to prove the inductive step, assume that for some $T \in [T_0]$, the statement in Lemma D.1 holds for all $t \leq T - 1$, and consider $\forall i, j, i', j' \in V$ such that $\boldsymbol{c}_2^{(T)}((i,j)) = \boldsymbol{c'}_2^{(T)}((i',j'))$. By the update rule of 2-WL, this implies that

$$\boldsymbol{c}_2^{(T-1)}((i,j)) = \boldsymbol{c'}_2^{(T-1)}((i',j'))$$
$$\{\boldsymbol{c}_2^{(T-1)}((k,j)) : k \in V\} = \{\boldsymbol{c'}_2^{(T-1)}((k,j')) : k \in V\} \tag{6}$$
$$\{\boldsymbol{c}_2^{(T-1)}((i,k)) : k \in V\} = \{\boldsymbol{c'}_2^{(T-1)}((i',k)) : k \in V\}$$

The first condition, thanks to the inductive hypothesis, implies that $\tilde{h}_{i,j}^{(T-1)} = \tilde{h'}_{i',j'}^{(T-1)}$. In particular, if $i \neq j$, then we have

$$a_{i,j} = a'_{i',j'}$$
$$e_{i,j} = e'_{i',j'} \tag{7}$$

The third condition implies that $\exists$ a permutation on $V$, which we can call $\xi_{i,i'}$, such that $\forall k \in V$,

$$\boldsymbol{c}_2^{(T-1)}((i,k)) = \boldsymbol{c'}_2^{(T-1)}((i',\xi_{i,i'}(k)))$$

By the inductive hypothesis, there is $\forall k \in V$,

$$\tilde{h}_{i,k}^{(T-1)} = \tilde{h'}_{i',\xi_{i,i'}(k)}^{(T-1)}$$

and moreover, $\xi_{i,i'}(k) = i'$ if and only if $k = i$. For $k \neq i$, we thus have

$$h_i^{(T-1)} = h'^{(T-1)}_{i'}$$
$$h_k^{(T-1)} = h'^{(T-1)}_{\xi_{i,i'}(k)}$$
$$a_{i,k} = a'_{i',\xi_{i,i'}(k)}$$
$$e_{i,k} = e'_{i',\xi_{i,i'}(k)}$$

Now, looking at the update rule at the $T$th layer of the MPNN,

$$
\begin{aligned}
m_i^{(T)} &= \sum_{k \in \mathcal{N}(i)} M_T(h_i^{(T-1)}, h_k^{(T-1)}, e_{i,k}) \\
&= \sum_{k \in V} a_{i,k} \cdot M_T(h_i^{(T-1)}, h_k^{(T-1)}, e_{i,k}) \\
&= \sum_{k \in V} a'_{i', \xi_{i,i'}(k)} \cdot M_T(h'^{(T-1)}_{i'}, h'^{(T-1)}_{\xi_{i,i'}(k)}, e'_{i', \xi_{i,i'}(k)}) \\
&= \sum_{k' \in V} a'_{i', k'} \cdot M_T(h'^{(T-1)}_{i'}, h'^{(T-1)}_{k'}, e'_{i', k'}) \\
&= m'^{(T)}_{i'}
\end{aligned}
$$

where between the third and the fourth line we made the substitution $k' = \xi_{i,i'}(k)$. Therefore,

$$
\begin{aligned}
h_i^{(T)} &= U_t(h_i^{(T-1)}, m_i^T) \\
&= U_t(h'^{(T-1)}_{i'}, m'^T_{i'}) \\
&= h'^{(T)}_{i'}
\end{aligned}
$$

By the symmetry between $i$ and $j$, we can also show that $h_j^{(T)} = h'^{(T)}_{j'}$. Hence, together with 7, we can conclude that

$$
\tilde{h}_{i,j}^{(T)} = \tilde{h}'^{(T)}_{i',j'},
$$

which proves the lemma. $\qquad\square$

Thus, the second result of this lemma tells us that $\forall i, j \in V^2$, $\tilde{h}_{i,j}^{(T_0)} = \tilde{h}'^{(T_0)}_{\eta_0(i,j)}$. Moreover, by the first result, $\exists$ a permutation on $V$, which we can call $\tau_0$, such that $\forall i \in V$, $\eta((i,i)) = (\tau_0(i), \tau_0(i))$. Combining the two, we have that $\forall i \in V$, $h_i^{(T_0)} = h'^{(T_0)}_{\tau(i)}$, and hence

$$
\{h_i^{(T_0)} : i \in V\} = \{h'^{(T_0)}_{i'} : i' \in V\} \tag{8}
$$

Therefore, $\hat{y} = \hat{y}'$, meaning that the MPNN returns identical outputs on the two graphs. $\qquad\square$

## E    Proof of Theorem 3.3 ($2$-WL is unable to induced-subgraph-count patterns of $3$ or more nodes)

*Proof Intuition.* Given any connected pattern of at least 3 nodes, such as the one in the left of Figure 2, we can construct a pair of graphs, such as the pair in the center and the right of Figure 2. They that have different induced-subgraph-counts of the pattern, and we can show that 2-WL cannot distinguish them. but cannot be distinguished from each other by 2-WL. For instance, if we run 2-WL on the pair of graphs in Figure 2, then there will be $c_2^{(t)}((1,3)) = c'^{(t)}_2((1,3))$, $c_2^{(t)}((1,2)) = c'^{(t)}_2((1,6))$, $c_2^{(t)}((1,6)) = c'^{(t)}_2((1,2))$, and so on. We can in fact show that $\{c_2^{(t)}(s) : s \in V^2\} = \{c'^{(t)}_2(s) : s \in V^2\}, \forall t$, which implies that 2-WL cannot distinguish the two graphs.

*Proof.* Say $G^{[\mathrm{P}]} = (V^{[\mathrm{P}]}, E^{[\mathrm{P}]}, x^{[\mathrm{P}]}, e^{[\mathrm{P}]})$ is a connected pattern of $m$ nodes, where $m > 2$, and thus $V^{[\mathrm{P}]} = [m]$.

First, if $G^{[\mathrm{P}]}$ is not a clique, then by definition, there exists two distinct nodes $i, j \in V^{[\mathrm{P}]}$ such that $i$ and $j$ are not connected by an edge. Assume without loss of generality that $i = 1$ and $j = 2$. Now, construct two graphs $G^{[1]} = (V = [2m], E^{[1]}, x^{[1]}, e^{[1]})$, $G^{[2]} = (V = [2m], E^{[2]}, x^{[2]}, e^{[2]})$ both with $2m$ nodes. For $G^{[1]}$, let $E^{[1]} = \{(i,j) : i, j \le m, (i,j) \in E^{[\mathrm{P}]}\} \cup \{(i+m, j+m) : i, j \le m, (i,j) \in E^{[\mathrm{P}]}\} \cup \{(1,2), (2,1), (1+m, 2+m), (2+m, 1+m)\}$; $\forall i \le m, x_i^{[1]} = x_{i+m}^{[1]} = x_i^{[\mathrm{P}]}$; $\forall (i,j) \in E^{[\mathrm{P}]}, e_{i,j}^{[1]} = e_{i+m,j+m}^{[1]} = e_{i,j}^{[\mathrm{P}]}$, and moreover we can randomly choose a value of edge

feature for $e^{[1]}_{1,2} = e^{[1]}_{2,1} = e^{[1]}_{1+m,2+m} = e^{[1]}_{2+m,1+m}$. For $G^{[2]}$, let $E^{[2]} = \{(i,j) : i,j \leq m, (i,j) \in E^{[P]}\} \cup \{(i+m, j+m) : i,j \leq m, (i,j) \in E^{[P]}\} \cup \{(1, 2+m), (2+m, 1), (1+m, 2), (2, 1+m)\}$; $\forall i \leq m, x^{[2]}_i = x^{[2]}_{i+m} = x^{[P]}_i$; $\forall (i,j) \in E^{[P]}, e^{[2]}_{i,j+m} = e^{[2]}_{i+m,j} = e^{[P]}_{i,j}$, and moreover we let $e^{[2]}_{1,2+m} = e^{[2]}_{2+m,1} = e^{[2]}_{1+m,2} = e^{[2]}_{2,1+m} = e^{[1]}_{1,2}$. In words, both $G^{[1]}$ and $G^{[2]}$ are constructed based on two copies of $G^{[P]}$, and the difference is that, $G^{[1]}$ adds the edges $\{(1,2), (2,1), (1+m, 2+m), (2+m, 1+m)\}$, whereas $G^{[2]}$ adds the edges $\{(1, 2+m), (2+m, 1), (1+m, 2), (2, 1+m)\}$, all with the same edge feature.

On one hand, by construction, 2-WL will not be able to distinguish $G^{[1]}$ from $G^{[2]}$. This is intuitive if we compare the rooted subtrees in the two graphs, as there exists a bijection from $V^{[1]}$ to $V^{[2]}$ that preserves the rooted subtree structure. A rigorous proof is given at the end of this section. In addition, we note that this is also consequence of the direct proof of Corollary 4.4 given in Appendix J, in which we will show that the same pair of graphs cannot be distinguished by 2-IGNs. Since 2-IGNs are no less powerful than 2-WL [39], this implies that 2-WL cannot distinguish them either.

On the other hand, $G^{[1]}$ and $G^{[2]}$ has different matching-count of the pattern. $G^{[1]}$ contains no subgraph isomorphic to $G^{[P]}$. Intuitively this is obvious; to be rigorous, note that firstly, neither the subgraph induced by the nodes $\{1, ..., m\}$ nor the subgraph induced by the nodes $\{1+m, ..., 2m\}$ is isomorphic to $G^{[P]}$, and secondly, the subgraph induced by any other set of $m$ nodes is not connected, whereas $G^{[P]}$ is connected. $G^{[2]}$, however, has at least two induced subgraphs isomorphic to $G^{[P]}$, one induced by the nodes $\{1, ..., m\}$, and the other induced by the nodes $\{1+m, ..., 2m\}$.

If $G^{[P]}$ is a clique, then we also first construct $G^{[1]}$, $G^{[2]}$ from $G^{[P]}$ as two copies of $G^{[P]}$. Then, for $G^{[1]}$, we pick two distinct nodes $1, 2 \in V^{[P]}$ and remove the edges $(1,2), (2,1), (1+m, 2+m)$ and $(2+m, 1+m)$ from $V^{[1]}$, while adding edges $(1, 2+m), (2+m, 1), (1+m, 2), (2, 1+m)$ with the same edge features. Then, $G^{[1]}$ contains no subgraph isomorphic to $G^{[P]}$, while $G^{[2]}$ contains two. Note that the pair of graphs is the same as the counterexample pair of graphs that could have been constructed in the non-clique case for the pattern that is a clique with one edge deleted. Hence 2-WL still can't distinguish $G^{[1]}$ from $G^{[2]}$.

$\square$

*Proof of* 2-WL failing to distinguish $G^{[1]}$ and $G^{[2]}$ :

To show that 2-WL cannot distinguish $G^{[1]}$ from $G^{[2]}$, we need to show that if we run 2-WL on the two graphs, then $\forall T, \{\boldsymbol{c}^{(T)}((i,j)) : i,j \in V\} = \{\boldsymbol{c}'^{(T)}((i,j)) : i,j \in V\}$. For this to hold, it is sufficient to find a bijective map $\eta : V^2 \to V^2$ such that $\boldsymbol{c}^{(T)}((i,j)) = \boldsymbol{c}'^{(T)}(\eta((i,j))), \forall i,j \in V$. First, we define a set $S = \{(1,2), (2,1), (1+m, 2+m), (2+m, 1+m), (1, 2+m), (2+m, 1), (1+m, 2), (2, 1+m)\}$, which represents the "special" pairs of nodes that capture the difference between $G^{[1]}$ and $G^{[2]}$. Then we can define $\eta : V^2 \to V^2$ as

$$\eta((i,j)) = \begin{cases} (i,j), & \text{if } (i,j) \notin S \\ (i, \text{MOD}_{2m}(j+m)), & \text{if } (i,j) \in S \end{cases}$$

Note that $\eta$ is a bijective. It is easy to verify that $\eta$ is a color-preserving map between node pairs in $G^{[1]}$ and node pairs in $G^{[2]}$ at initialization, i.e. $\boldsymbol{c}^{(0)}((i,j)) = \boldsymbol{c}'^{(0)}(\eta((i,j))), \forall i,j \in V$. We will prove by induction that in fact it remains such a color-preserving map at any iteration $T$. The inductive step that we need to prove is,

**Lemma E.1.** *For any positive integer t, supposing that* $\boldsymbol{c}^{(t-1)}((i,j)) = \boldsymbol{c}'^{(t-1)}(\eta((i,j))), \forall i,j \in V$, *then we also have* $\boldsymbol{c}^{(t)}((i,j)) = \boldsymbol{c}'^{(t)}(\eta((i,j))), \forall i,j \in V$.

*Proof of Lemma E.1:* By the update rule of 2-WL, $\forall i,j \in V$, to show that $\boldsymbol{c}^{(t)}((i,j)) = \boldsymbol{c}'^{(t)}(\eta((i,j)))$, we need to establish three conditions:

$$\boldsymbol{c}^{(t-1)}((i,j)) = \boldsymbol{c}'^{(t-1)}(\eta((i,j))) \tag{9}$$

$$\{\boldsymbol{c}^{(t-1)}(\tilde{s}) : \tilde{s} \in N_1((i,j))\} = \{\boldsymbol{c}'^{(t-1)}(\tilde{s}) : \tilde{s} \in N_1(\eta((i,j)))\} \qquad (10)$$

$$\{\boldsymbol{c}^{(t-1)}(\tilde{s}) : \tilde{s} \in N_2((i,j))\} = \{\boldsymbol{c}'^{(t-1)}(\tilde{s}) : \tilde{s} \in N_2(\eta((i,j)))\} \qquad (11)$$

The first condition is already guaranteed by the inductive hypothesis. Now we prove the last two conditions by examining different cases separately below.

Case 1 $i, j \notin \{1, 2, 1+m, 2+m\}$
Then $\eta((i,j)) = (i,j)$, and $N_1((i,j)) \cap S = \emptyset$, $N_2((i,j)) \cap S = \emptyset$. Therefore, $\eta$ restricted to $N_1((i,j))$ or $N_2((i,j))$ is the identity map, and thus

$$\{\boldsymbol{c}^{(t-1)}(\tilde{s}) : \tilde{s} \in N_1((i,j))\} = \{\boldsymbol{c}'^{(t-1)}(\eta(\tilde{s})) : \tilde{s} \in N_1((i,j))\}$$
$$= \{\boldsymbol{c}'^{(t-1)}(\tilde{s}) : \tilde{s} \in N_1(\eta((i,j)))\},$$

thanks to the inductive hypothesis. Similar for the condition (11).

Case 2 $i \in \{1, 1+m\}, j \notin \{1, 2, 1+m, 2+m\}$
Then $\eta((i,j)) = (i,j)$, $N_2((i,j)) \cap S = \{(i,2), (i,2+m)\}$, and $N_1((i,j)) \cap S = \emptyset$. To show condition (11), note that $\eta$ is the identity map when restricted to $N_2((i,j)) \setminus \{(i,2), (i,2+m)\}$, and hence

$$\{\boldsymbol{c}^{(t-1)}(\tilde{s}) : \tilde{s} \in N_2((i,j)) \setminus \{(i,2), (i,2+m)\}\} = \{\boldsymbol{c}'^{(t-1)}(\tilde{s}) : \tilde{s} \in N_2((i,j)) \setminus \{(i,2), (i,2+m)\}\}$$

Moreover, $\eta((i,2)) = (i, 2+m)$ and $\eta((i, 2+m)) = (i,2)$. Hence, by the inductive hypothesis, $\boldsymbol{c}^{(t-1)}((i,2)) = \boldsymbol{c}'^{(t-1)}((i, 2+m))$ and $\boldsymbol{c}^{(t-1)}((i, 2+m)) = \boldsymbol{c}'^{(t-1)}((i,2))$. Therefore,

$$\{\boldsymbol{c}^{(t-1)}(\tilde{s}) : \tilde{s} \in N_2((i,j))\} = \{\boldsymbol{c}'^{(t-1)}(\tilde{s}) : \tilde{s} \in N_2((i,j))\}$$
$$= \{\boldsymbol{c}'^{(t-1)}(\tilde{s}) : \tilde{s} \in N_2(\eta((i,j)))\},$$

which shows condition (11). Condition (10) is easily seen as $\eta$ restricted to $N_1((i,j))$ is the identity map.

Case 3 $j \in \{1, 1+m\}, i \notin \{1, 2, 1+m, 2+m\}$
There is $\eta((i,j)) = (i,j)$, $N_1((i,j)) \cap S = \{(2,j), (2+m,j)\}$, and $N_2((i,j)) \cap S = \emptyset$. Hence the proof can be carried out analogously to case 2.

Case 4 $i \in \{2, 2+m\}, j \notin \{1, 2, 1+m, 2+m\}$
There is $\eta((i,j)) = (i,j)$, $N_2((i,j)) \cap S = \{(i,1), (i,1+m)\}$, and $N_1((i,j)) \cap S = \emptyset$. Hence the proof can be carried out analogously to case 2.

Case 5 $j \in \{2, 2+m\}, i \notin \{1, 2, 1+m, 2+m\}$
There is $\eta((i,j)) = (i,j)$, $N_1((i,j)) \cap S = \{(1,j), (1+m,j)\}$, and $N_2((i,j)) \cap S = \emptyset$. Hence the proof can be carried out analogously to case 2.

Case 6 $(i,j) \in S$
There is $\eta((i,j)) = (i, \text{MOD}_{2m}(j))$, $N_1((i,j)) \cap S = \{(i,j), (\text{MOD}_{2m}(i), j)\}$, $N_2((i,j)) \cap S = \{(i,j), (i, \text{MOD}_{2m}(j))\}$. Thus, $N_1(\eta((i,j))) = N_1((i, \text{MOD}_{2m}(j)))$, $N_2(\eta((i,j))) = N_2((i, \text{MOD}_{2m}(j))) = N_2((i,j))$. Once again, $\eta$ is the identity map when restricted to $N_1((i,j)) \setminus S$ or $N_2((i,j)) \setminus S$. Hence, by the inductive hypothesis, there is

$$\{\boldsymbol{c}^{(t-1)}(\tilde{s}) : \tilde{s} \in N_1((i,j)) \setminus \{(i,j), (\text{MOD}_{2m}(i), j)\}\} = \{\boldsymbol{c}'^{(t-1)}(\tilde{s}) : \tilde{s} \in N_1((i,j)) \setminus \{(i,j), (\text{MOD}_{2m}(i), j)\}\}$$

$$\{\boldsymbol{c}^{(t-1)}(\tilde{s}) : \tilde{s} \in N_2((i,j)) \setminus \{(i,j), (i, \text{MOD}_{2m}(j))\}\} = \{\boldsymbol{c}'^{(t-1)}(\tilde{s}) : \tilde{s} \in N_2((i,j)) \setminus \{(i,j), (i, \text{MOD}_{2m}(j))\}\}$$

Also from the inductive hypothesis, we have

$$\boldsymbol{c}^{(t-1)}((i,j)) = \boldsymbol{c}'^{(t-1)}(\eta((i,j)))$$
$$= \boldsymbol{c}'^{(t-1)}((i, \text{MOD}_{2m}(j))), \qquad (12)$$

$$\begin{aligned}
\boldsymbol{c}^{(t-1)}((i,j)) &= \boldsymbol{c}^{(t-1)}((j,i)) \\
&= \boldsymbol{c}'^{(t-1)}(\eta((j,i))) \\
&= \boldsymbol{c}'^{(t-1)}((j,\text{MOD}_{2m}(i))) \\
&= \boldsymbol{c}'^{(t-1)}((\text{MOD}_{2m}(i),j)),
\end{aligned} \tag{13}$$

$$\begin{aligned}
\boldsymbol{c}^{(t-1)}((i,\text{MOD}_{2m}(j))) &= \boldsymbol{c}'^{(t-1)}(\eta((i,\text{MOD}_{2m}(j)))) \\
&= \boldsymbol{c}'^{(t-1)}((i,\text{MOD}_{2m}(\text{MOD}_{2m}(j)))) \\
&= \boldsymbol{c}'^{(t-1)}((i,j)),
\end{aligned} \tag{14}$$

$$\begin{aligned}
\boldsymbol{c}^{(t-1)}((\text{MOD}_{2m}(i),j)) &= \boldsymbol{c}^{(t-1)}((j,\text{MOD}_{2m}(i))) \\
&= \boldsymbol{c}'^{(t-1)}(\eta((j,\text{MOD}_{2m}(i)))) \\
&= \boldsymbol{c}'^{(t-1)}((j,\text{MOD}_{2m}(\text{MOD}_{2m}(i)))) \\
&= \boldsymbol{c}'^{(t-1)}((j,i)) \\
&= \boldsymbol{c}'^{(t-1)}((i,j)),
\end{aligned} \tag{15}$$

where in (13) and (15), the first and the last equalities are thanks to the symmetry of the coloring between any pair of nodes $(i',j')$ and its "reversed" version $(j',i')$, which persists throughout all iterations, as well as the fact that if $(i',j') \in S$, then $(j',i') \in S$. Therefore, we now have

$$\{\boldsymbol{c}^{(t-1)}(\tilde{s}) : \tilde{s} \in N_1((i,j))\} = \{\boldsymbol{c}'^{(t-1)}(\tilde{s}) : \tilde{s} \in N_1((i,j))\} \tag{16}$$

$$\{\boldsymbol{c}^{(t-1)}(\tilde{s}) : \tilde{s} \in N_2((i,j))\} = \{\boldsymbol{c}'^{(t-1)}(\tilde{s}) : \tilde{s} \in N_2((i,j))\} \tag{17}$$

Since $\eta((i,j)) = (i,\text{MOD}_{2m}(j))$, we have

$$\begin{aligned}
N_1(\eta((i,j))) &= \{(k,\text{MOD}_{2m}(j)) : k \in V\} \\
&= \{(k,\text{MOD}_{2m}(j)) : (\text{MOD}_{2m}(k),j) \in N_1((i,j))\} \\
&= \{(\text{MOD}_{2m}(k),\text{MOD}_{2m}(j)) : (k,j) \in N_1((i,j))\}
\end{aligned}$$

Thanks to the symmetry of the coloring under the map $(i',j') \to (\text{MOD}_{2m}(i'),\text{MOD}_{2m}(j'))$, we then have

$$\begin{aligned}
\{\boldsymbol{c}'^{(t-1)}(\tilde{s}) : \tilde{s} \in N_1(\eta((i,j)))\} &= \{\boldsymbol{c}'^{(t-1)}((\text{MOD}_{2m}(k),\text{MOD}_{2m}(j))) : (k,j) \in N_1((i,j))\} \\
&= \{\boldsymbol{c}'^{(t-1)}((k,j)) : (k,j) \in N_1((i,j))\} \\
&= \{\boldsymbol{c}'^{(t-1)}(\tilde{s}) : \tilde{s} \in N_1((i,j))\}
\end{aligned}$$

Therefore, combined with (16), we see that (10) is proved. (11) is a straightforward consequence of (17), since $N_2((i,j)) = N_2(\eta((i,j)))$.

**Case 7** $i,j \in \{1,1+m\}$
There is $\eta((i,j)) = (i,j)$, $N_2((i,j)) \cap S = \{(i,2),(i,2+m)\}$, and $N_1((i,j)) \cap S = \{(2,j),(2+m,j)\}$. Thus, both (10) and (11) can be proved analogously to how (11) is proved for case 2.

**Case 8** $i,j \in \{2,2+m\}$
There is $\eta((i,j)) = (i,j)$, $N_2((i,j)) \cap S = \{(i,1),(i,1+m)\}$, and $N_1((i,j)) \cap S = \{(1,j),(1+m,j)\}$. Thus, both (10) and (11) can be proved analogously to how (11) is proved for case 2.

With conditions (10) and (11) shown for all pairs of $(i,j) \in V^2$, we know that by the update rules of 2-WL, there is $\boldsymbol{c}^{(t)}((i,j)) = \boldsymbol{c}'^{(t)}(\eta((i,j))), \forall i,j \in V$.

$\square$

With Lemma E.1 justifying the inductive step, we see that for any positive integer $T$, there is $\boldsymbol{c}^{(T)}((i,j)) = \boldsymbol{c}'^{(T)}(\eta((i,j))), \forall i, j \in V$. Hence, we can conclude that $\forall T, \{\boldsymbol{c}^{(T)}((i,j)) : i, j \in V\} = \{\boldsymbol{c}'^{(T)}((i,j)) : i, j \in V\}$, which implies that the two graphs cannot be distinguished by 2-WL.

$\square$

## F   Proof of Theorem 3.5 (MPNNs are able to subgraph-count star-shaped patterns)

(See Section 2.1 of Arvind et al. [2] for a proof for the case where all nodes have identical features.)

*Proof.* Without loss of generality, we represent a star-shaped pattern by $G^{[\mathrm{P}]} = (V^{[\mathrm{P}]}, E^{[\mathrm{P}]}, x^{[\mathrm{P}]}, e^{[\mathrm{P}]})$, where $V^{[\mathrm{P}]} = [m]$ (with node 1 representing the center) and $E^{[\mathrm{P}]} = \{(1,i) : 2 \le i \le m\} \cup \{(i,1) : 2 \le i \le m\}$.

Given a graph $G$, for each of its node $j$, we define $N(j)$ as the set of its neighbors in the graph. Then the neighborhood centered at $j$ contributes to $\mathsf{C}_S(G, G^{[\mathrm{P}]})$ if and only if $x_j = x_1^{[\mathrm{P}]}$ and $\exists S \subseteq N(j)$ such that the multiset $\{(x_k, e_{jk}) : k \in S\}$ equals the multiset $\{(x_k^{[\mathrm{P}]}, e_{1k}^{[\mathrm{P}]}) : 2 \le k \le m\}$. Moreover, the contribution to the number $\mathsf{C}_S(G, G^{[\mathrm{P}]})$ equals the number of all such subsets $S \subseteq N(j)$. Hence, we have the following decomposition

$$\mathsf{C}_S(G, G^{[\mathrm{P}]}) = \sum_{j \in V} f^{[\mathrm{P}]}\Big(x_j, \{(x_k, e_{jk}) : k \in N(j)\}\Big),$$

where $f^{[\mathrm{P}]}$, is defined for every 2-tuple consisting of a node feature and a multiset of pairs of node feature and edge feature (i.e., objects of the form

$$\Big(x, M = \{(x_\alpha, e_\alpha) : \alpha \in K\}\Big)$$

where $K$ is a finite set of indices) as

$$f^{[\mathrm{P}]}(x, M) = \begin{cases} 0 & \text{if } x \ne x_1^{[\mathrm{P}]} \\ \#_M^{[\mathrm{P}]} & \text{if } x = x_1^{[\mathrm{P}]} \end{cases}$$

where $\#_M^{[\mathrm{P}]}$ denotes the number of sub-multisets of $M$ that equals the multiset $\{(x_k^{[\mathrm{P}]}, e_{1k}^{[\mathrm{P}]}) : 2 \le k \le m\}$.

Thanks to Corollary 6 of Xu et al. [64] based on Zaheer et al. [72], we know that $f^{[\mathrm{P}]}$ can be expressed by some message-passing function in an MPNN. Thus, together with summation as the readout function, MPNN is able to express $\mathsf{C}_S(G, G^{[\mathrm{P}]})$.   $\square$

## G   Proof of Theorem 3.7 ($k$-WL is able to count patterns of $k$ or fewer nodes)

*Proof.* Suppose we run $k$-WL on two graphs, $G^{[1]}$ and $G^{[2]}$. In $k$-WL, the colorings of the $k$-tuples are initialized according to their isomorphism types as defined in Appendix C. Thus, if for some pattern of no more than $k$ nodes, $G^{[1]}$ and $G^{[2]}$ have different matching-count or containment-count, then there exists an isomorphism type of $k$-tuples such that $G^{[1]}$ and $G^{[2]}$ differ in the number of $k$-tuples under this type. This implies that $\{\boldsymbol{c}_k^{(0)}(s) : s \in (V^{[1]})^k\} \ne \{\boldsymbol{c}'_k^{(0)}(s') : s' \in (V^{[2]})^k\}$, and hence the two graphs can be distinguished at the 0th iteration of $k$-WL.   $\square$

## H   Proof of Theorem 3.9 ($T$ iterations of $k$-WL cannot induced-subgraph-count path patterns of size $(k+1)2^T$ or more)

*Proof.* For any integer $m \ge (k+1)2^T$, we will construct two graphs $G^{[1]} = (V^{[1]} = [2m], E^{[1]}, x^{[1]}, e^{[1]})$ and $G^{[2]} = (V^{[2]} = [2m], E^{[2]}, x^{[2]}, e^{[2]})$, both with $2m$ nodes but with

different matching-counts of $H_m$, and show that $k$-WL cannot distinguish them. Define $E_{double} = \{(i, i+1) : 1 \le i < m\} \cup \{(i+1, i) : 1 \le i < m\} \cup \{(i+m, i+m+1) : 1 \le i < m\} \cup \{(i+m+1, i+m) : 1 \le i < m\}$, which is the edge set of a graph that is exactly two disconnected copies of $H_m$. For $G^{[1]}$, let $E^{[1]} = E_{double} \cup \{(1, m), (m, 1), (1+m, 2m), (2m, 1+m)\}$; $\forall i \le m, x_i^{[1]} = x_{i+m}^{[1]} = x_i^{[\mathrm{H_m}]}$; $\forall (i, j) \in E^{[\mathrm{H_m}]}, e_{i,j}^{[1]} = e_{j,i}^{[1]} = e_{i+m,j+m}^{[1]} = e_{j+m,i+m}^{[1]} = e_{i,j}^{[\mathrm{H_m}]}$, and moreover, we can randomly choose a value of edge feature for $e_{1,m}^{[1]} = e_{m,1}^{[1]} = e_{1+m,2m}^{[1]} = e_{2m,1+m}^{[1]}$. For $G^{[2]}$, let $E^{[2]} = E_{double} \cup \{(1, 2m), (2m, 1), (m, 1+m), (1+m, 2m)\}$; $\forall i \le m, x_i^{[2]} = x_{i+m}^{[2]} = x_i^{[\mathrm{H_m}]}$; $\forall (i, j) \in E^{[\mathrm{H_m}]}, e_{i,j}^{[1]} = e_{j,i}^{[1]} = e_{i+m,j+m}^{[1]} = e_{j+m,i+m}^{[1]} = e_{i,j}^{[\mathrm{H_m}]}$, and moreover, set $e_{1,2m}^{[2]} = e_{2m,1}^{[2]} = e_{m,1+m}^{[2]} = e_{1+m,m}^{[2]} = e_{1,m}^{[1]}$. In words, both $G^{[1]}$ and $G^{[2]}$ are constructed based on two copies of $H_m$, and the difference is that, $G^{[1]}$ adds the edges $\{(1, m), (m, 1), (1+m, 2m), (2m, 1+m)\}$, whereas $G^{[2]}$ adds the edges $\{(1, 2m), (2m, 1), (m, 1+m), (1+m, m)\}$, all with the same edge feature. For the case $k = 3, m = 8, T = 1$, for example, the constructed graphs are illustrated in Figure 4.

Can $G^{[1]}$ and $G^{[2]}$ be distinguished by $k$-WL? Let $c_k^{(t)}, c'_k^{(t)}$ be the coloring functions of $k$-tuples for $G^{[1]}$ and $G^{[2]}$, respectively, obtained after running $k$-WL on the two graphs simultaneously for $t$ iterations. To show that the answer is negative, we want to prove that

$$\{c_k^{(T)}(s) : s \in [2m]^k\} = \{c'_k{}^{(T)}(s) : s \in [2m]^k\} \tag{18}$$

To show this, if is sufficient to find a permutation $\eta : [2m]^k \to [2m]^k$ such that $\forall$ $k$-tuple $s \in [2m]^k, c_k^{(T)}(s) = c'_k{}^{(T)}(\eta(s))$. Before defining such an $\eta$, we need the following lemma.

**Lemma H.1.** *Let $p$ be a positive integer. If $m \ge (k+1)p$, then $\forall s \in [2m]^k, \exists i \in [m]$ such that $\{i, i+1, ..., i+p-1\} \cap \{\mathrm{MOD}_m(j) : j \in s\} = \emptyset$.*

*Proof of Lemma H.1:* We can use a simple counting argument to show this. For $u \in [k+1]$, define $A_u = \{up, up+1, ..., (u+1)p-1\} \cup \{up+m, up+1+m, ..., (u+1)p-1+m\}$. Then $|A_u| = 2p$, $A_u \cap A_{u'} = \emptyset$ if $u \ne u'$, and

$$[2m] \supseteq \bigcup_{u \in [k+1]} A_u, \tag{19}$$

since $m \ge (k+1)p$. Suppose that the claim is not true, then each $A_i$ contains at least one node in $s$, and therefore

$$s \supseteq (s \cap [2m]) \supseteq \bigcup_{u \in [k+1]} (s \cap A_u),$$

which contains at least $k+1$ nodes, which is contradictory. $\qquad\square$

With this lemma, we see that $\forall s \in [2m]^k, \exists i \in [m]$ such that $\forall j \in s, \mathrm{MOD}_m(j)$ either $< i$ or $\ge i + 2^{T+1} - 1$. Thus, we can first define the mapping $\chi : [2m]^k \to [m]$ from a $k$-tuple $s$ to the smallest such node index $i \in [m]$. Next, $\forall i \in [m]$, we define a mapping $\tau_i$ from $[2m]$ to $[2m]$ as

$$\tau_i(j) = \begin{cases} j, & \text{if } \mathrm{MOD}_m(j) \le i \\ \mathrm{MOD}_{2m}(j+m), & \text{otherwise} \end{cases} \tag{20}$$

$\tau_i$ is a permutation on $[2m]$. For $\forall i \in [m]$, this allows us to define a mapping $\zeta_i$ from $[2m]^k \to [2m]^k$ as, $\forall s = (i_1, ..., i_k) \in [2m]^k$,

$$\zeta_i(s) = (\tau_i(i_1), ..., \tau_i(i_k)). \tag{21}$$

Finally, we define a mapping $\eta$ from $[2m]^k \to [2m]^k$ as,

$$\eta(s) = \zeta_{\chi(s)}(s) \tag{22}$$

The maps $\chi, \tau$ and $\eta$ are illustrated in Figure 4.

$$G^{[1]} \qquad\qquad\qquad\qquad G^{[2]}$$

Figure 4: Illustration of the construction in the proof of Theorem 3.9 in Appendix H. In this particular case, $k = 3$, $m = 8$, $T = 1$. If we consider $s = (1, 12, 8)$ as an example, where the corresponding nodes are marked by blue squares in $G^{[1]}$, there is $\chi(s) = 2$, and thus $\eta(s) = \zeta_2(s) = (1, 4, 16)$, which are marked by blue squares in $G^{[2]}$. Similarly, if we consider $s = (3, 14, 15)$, then $\chi(s) = 4$, and thus $\eta(s) = \zeta_4(s) = (3, 6, 7)$. In both cases, we see that the isomorphism type of $s$ in $G^{[1]}$ equals the isomorphism type of $\eta(s)$ in $G^{[2]}$. In the end, we will show that $\boldsymbol{c}_k^{(T)}(s) = \boldsymbol{c}_k'^{(T)}(\eta(s))$.

To fulfill the proof, there are two things we need to show about $\eta$. First, we want it to be a permutation on $[2m]^k$. To see this, observe that $\chi(s) = \chi(\eta(s))$, and hence $\forall s \in [2m]^k, (\eta \circ \eta)(s) = (\zeta_{\chi(\eta(s))} \circ \zeta_{\chi(s)})(s) = s$, since $\forall i \in [m], \tau_i \circ \tau_i$ is the identity map on $[2m]$.

Second, we need to show that $\forall s \in [2m]^k, \boldsymbol{c}_k^{(T)}(s) = \boldsymbol{c}_k'^{(T)}(\eta(s))$. This will be a consequence of the following lemma.

**Lemma H.2.** *At iteration $t$, $\forall s \in [2m]^k$, $\forall i$ such that $\forall j \in s$, either $\mathrm{MOD}_m(j) < i$ or $\mathrm{MOD}_m(j) \geq i + 2^t$, there is*

$$\boldsymbol{c}_k^{(t)}(s) = \boldsymbol{c}_k'^{(t)}(\zeta_i(s)) \tag{23}$$

*Remark: This statement allows $i$ to depend on $s$, as will be the case when we apply this lemma to $\eta(s) = \zeta_{\chi(s)}(s)$, where we set $i$ to be $\chi(s)$.*

*Proof of Lemma H.2:* Notation-wise, for any $k$-tuple, $s = (i_1, ..., i_k)$, and for $w \in [k]$, use $\mathrm{I}_w(s)$ to denote the $w$th entry of $s$, $i_w$.

The lemma can be shown by using induction on $t$. Before looking at the base case $t = 0$, we will first show the inductive step, which is:

$$\forall \bar{T}, \text{ suppose the lemma holds for all } t \leq \bar{T} - 1,$$
$$\text{then it also holds for } t = \bar{T}. \tag{24}$$

*Inductive step*:
Fix a $\bar{T}$ and suppose the lemma holds for all $t \leq \bar{T} - 1$. Under the condition that $\forall j \in s$, either $\mathrm{MOD}_m(j) < i$ or $\mathrm{MOD}_m(j) \geq i + 2^{\bar{T}}$, to show $\boldsymbol{c}_k^{(\bar{T})}(s) = \boldsymbol{c}_k'^{(\bar{T})}(\zeta_i(s))$, we need two things to hold:

1. $\boldsymbol{c}_k^{(\bar{T}-1)}(s) = \boldsymbol{c}_k'^{(\bar{T}-1)}(\zeta_i(s))$

2. $\forall w \in [k], \{\boldsymbol{c}_k^{(\bar{T}-1)}(\tilde{s}) : \tilde{s} \in N_w(s)\} = \{\boldsymbol{c}_k'^{(\bar{T}-1)}(\tilde{s}) : \tilde{s} \in N_w(\zeta_i(s))\}$

The first condition is a consequence of the inductive hypothesis, as $i + 2^{\bar{T}} > i + 2^{(\bar{T}-1)}$. For the second condition, it is sufficient to find for all $w \in [k]$, a bijective mapping $\xi$ from $N_w(s)$ to $N_w(\zeta_i(s))$ such that $\forall \tilde{s} \in N_w(s), \boldsymbol{c}_k^{(\bar{T}-1)}(\tilde{s}) = \boldsymbol{c}_k'^{(\bar{T}-1)}(\xi(\tilde{s}))$.

We then define $\beta(i, \tilde{s}) =$

$$\begin{cases} \text{MOD}_m(\mathbf{I}_w(\tilde{s})) + 1, & \text{if } i \leq \text{MOD}_m(\mathbf{I}_w(\tilde{s})) < i + 2^{\bar{T}-1} \\ i, & \text{otherwise} \end{cases} \tag{25}$$

Now, consider any $\tilde{s} \in N_w(s)$. Note that $\tilde{s}$ and $s$ differ only in the $w$th entry of the $k$-tuple.

- If $i \leq \text{MOD}_m(\mathbf{I}_w(\tilde{s})) < i + 2^{\bar{T}-1}$, then $\forall j \in \tilde{s}$,
    - either $j \in s$, in which case either $\text{MOD}_m(j) < i < \text{MOD}_m(\mathbf{I}_w(\tilde{s})) + 1 = \beta(i, \tilde{s})$ or $\text{MOD}_m(j) \geq i + 2^{\bar{T}} \geq \text{MOD}_m(\mathbf{I}_w(\tilde{s})) + 1 + 2^{\bar{T}-1} = \beta(i, \tilde{s}) + 2^{\bar{T}-1}$,
    - or $j = \mathbf{I}_w(\tilde{s})$, in which case $\text{MOD}_m(j) < \text{MOD}_m(\mathbf{I}_w(\tilde{s})) + 1 = \beta(i, \tilde{s})$.

- If $\text{MOD}_m(\mathbf{I}_w(\tilde{s})) < i$ or $\text{MOD}_m(\mathbf{I}_w(\tilde{s})) \geq i + 2^{\bar{T}-1}$, then $\forall j \in \tilde{s}$,
    - either $j \in s$, in which case either $\text{MOD}_m(j) < i = \beta(i, \tilde{s})$ or $\text{MOD}_m(j) \geq i + 2^{\bar{T}} \geq \beta(i, \tilde{s}) + 2^{\bar{T}-1}$,
    - or $j = \mathbf{I}_w(\tilde{s})$, in which case either $\text{MOD}_m(j) < i = \beta(i, \tilde{s})$ or $\text{MOD}_m(j) \geq i + 2^{\bar{T}-1} \geq \beta(i, \tilde{s}) + 2^{\bar{T}-1}$.

Thus, in all cases, there is $\forall j \in \tilde{s}$, either $\text{MOD}_m(j) < \beta(i, \tilde{s})$, or $\text{MOD}_m(j) \geq i + 2^{\bar{T}-1}$. Hence, by the inductive hypothesis, we have $\boldsymbol{c}_k^{(\bar{T}-1)}(\tilde{s}) = \boldsymbol{c}'_k^{(\bar{T}-1)}(\zeta_{\beta(i,\tilde{s})}(\tilde{s}))$. This inspires us to define, for $\forall w \in [k], \forall \tilde{s} \in N_w(s)$,

$$\xi(\tilde{s}) = \zeta_{\beta(i,\tilde{s})}(\tilde{s}) \tag{26}$$

Additionally, we still need to prove that, firstly, $\xi$ maps $N_w(s)$ to $N_w(\zeta_i(s))$, and secondly, $\xi$ is a bijection. For the first statement, note that $\forall \tilde{s} \in N_w(s)$, $\zeta_{\beta(i,\tilde{s})}(s) = \zeta_i(s)$ because $s$ contains no entry between $i$ and $\beta(i, \tilde{s})$, with the latter being less than $i + 2^{\bar{T}}$. Hence, if $\tilde{s} \in N_w(s)$, then $\forall w' \in [k]$ with $w' \neq w$, there is $\mathbf{I}_{w'}(\tilde{s}) = \mathbf{I}_{w'}(s)$, and therefore $\mathbf{I}_{w'}(\xi(\tilde{s})) = \mathbf{I}_{w'}(\zeta_{\beta(i,\tilde{s})}(\tilde{s})) = \tau_{\beta(i,\tilde{s})}(\mathbf{I}_{w'}(\tilde{s})) = \tau_{\beta(i,\tilde{s})}(\mathbf{I}_{w'}(s)) = \mathbf{I}_{w'}(\zeta_{\beta(i,\tilde{s})}(s)) = \mathbf{I}_{w'}(\zeta_i(s))$, which ultimately implies that $\xi(\tilde{s}) \in N_w(\zeta_i(s))$.

For the second statement, note that since $\mathbf{I}_w(\xi(\tilde{s})) = \tau_{\beta(i,\tilde{s})}(\mathbf{I}_w(\tilde{s}))$ (by the definition of $\zeta$), there is $\text{MOD}_m(\mathbf{I}_w(\xi(\tilde{s}))) = \text{MOD}_m(\tau_{\beta(i,\tilde{s})}(\mathbf{I}_w(\tilde{s}))) = \text{MOD}_m(\mathbf{I}_w(\tilde{s}))$, and therefore $\beta(i, \xi(\tilde{s})) = \beta(i, \tilde{s})$. Thus, we know that $(\xi \circ \xi)(\tilde{s}) = (\zeta_{\beta(i,\xi(\tilde{s}))} \circ \zeta_{\beta(i,\tilde{s})})(\tilde{s}) = (\zeta_{\beta(i,\tilde{s})} \circ \zeta_{\beta(i,\tilde{s})})(\tilde{s}) = \tilde{s}$. This implies that $\xi$ is a bijection from $N_w(s)$ to $N_w(\zeta_i(s))$.

This concludes the proof of the inductive step.

*Base case*:
We need to show that

$$\forall s \in [2m]^k, \forall i^* \text{ such that } \forall j \in s, \text{ either } \text{MOD}_m(j) < i^*$$
$$\text{or } \text{MOD}_m(j) \geq i^* + 1, \text{ there is } \boldsymbol{c}_k^{(0)}(s) = \boldsymbol{c}'_k^{(0)}(\zeta_{i^*}(s)) \tag{27}$$

Due to the way in which the colorings of the $k$-tuples are initialized in $k$-WL, the statement above is equivalent to showing that $s$ in $G^{[1]}$ and $\zeta_{i^*}(s)$ in $G^{[2]}$ have the same isomorphism type, for which we need the following to hold.

**Lemma H.3.** *Say $s = (i_1, ..., i_k)$, in which case $\zeta_{i^*}(s) = (\tau_{i^*}(i_1), ..., \tau_{i^*}(i_k))$. Then*

1. $\forall i_\alpha, i_\beta \in s, i_\alpha = i_\beta \Leftrightarrow \tau_{i^*}(i_\alpha) = \tau_{i^*}(i_\beta)$

2. $\forall i_\alpha \in s, x_{i_\alpha}^{[1]} = x_{\tau_{i^*}(i_\alpha)}^{[2]}$

3. $\forall i_\alpha, i_\beta \in s, (i_\alpha, i_\beta) \in E^{[1]} \Leftrightarrow (\tau_{i^*}(i_\alpha), \tau_{i^*}(i_\beta)) \in E^{[2]}$, *and moreover, if either is true,* $e_{i_\alpha, i_\beta}^{[1]} = e_{\tau_{i^*}(i_\alpha), \tau_{i^*}(i_\beta)}^{[2]}$

*Proof of Lemma H.3:*

1. This is true since $\tau_{i^*}$ is a permutation on $[2m]$.

2. This is true because by the construction of the two graphs, $\forall i \in [2m], x_i^{[1]} = x_i^{[2]}$, and moreover $x_i^{[1]} = x_{i+m}^{[1]}$ if $i \leq m$.

3. Define $S = \{(1, m), (m, 1), (1 + m, 2m), (2m, 1 + m), (1, 2m), (2m, 1), (m, 1 + m), (1 + m, 2m)\}$, which is the set of "special" pairs of nodes in which $G^{[1]}$ and $G^{[2]}$ differ. Note that $\forall (i_\alpha, i_\beta) \in [2m]^2, (i_\alpha, i_\beta) \in S$ if and only if the sets $\{\text{MOD}_m(i_\alpha), \text{MOD}_m(i_\beta)\} = \{1, m\}$.

   By the assumption on $i^*$ in (27), we know that $i_\alpha, i_\beta \notin \{i^*, i^* + m\}$. Now we look at 16 different cases separately, which comes from 4 possibilities for each of $i_\alpha$ and $i_\beta$: $i_\alpha$ (or $i_\beta$) belonging to $\{1, ..., i^* - 1\}, \{i^* + 1, ..., m\}, \{1 + m, ..., i^* - 1 + m\}$, or $\{i^* + 1 + m, ..., 2m\}$

Case 1 $1 \leq i_\alpha, i_\beta < i^*$
   Then $\tau_{i^*}(i_\alpha) = i_\alpha, \tau_{i^*}(i_\beta) = i_\beta$. In addition, as $\text{MOD}_m(i_\alpha), \text{MOD}_m(i_\beta) \neq m$, there is $(i_\alpha, i_\beta) \notin S$. Thus, if $(i_\alpha, i_\beta) \in E^{[1]}$, then $(i_\alpha, i_\beta) \in E_{double} \subset E^{[2]}$, and moreover, $e_{i_\alpha, i_\beta}^{[1]} = e_{i_\alpha, i_\beta}^{[H_m]} = e_{i_\alpha, i_\beta}^{[2]} = e_{\tau_{i^*}(i_\alpha), \tau_{i^*}(i_\beta)}^{[2]}$. Same for the other direction.

Case 2 $1 + m \leq i_\alpha, i_\beta < i^* + m$
   Similar to case 1.

Case 3 $i^* + 1 \leq i_\alpha, i_\beta \leq m$
   Then $\tau_{i^*}(i_\alpha) = i_\alpha + m, \tau_{i^*}(i_\beta) = i_\beta + m$. In addition, as $\text{MOD}_m(i_\alpha), \text{MOD}_m(i_\beta) \neq 1$, there is $(i_\alpha, i_\beta) \notin S$. Thus, if $(i_\alpha, i_\beta) \in E^{[1]}$, then $(i_\alpha, i_\beta) \in E_{double}$, and hence $(i_\alpha + m, i_\beta + m) \in E_{double} \subset E^{[2]}$, and moreover, $e_{i_\alpha, i_\beta}^{[1]} = e_{i_\alpha, i_\beta}^{[H_m]} = e_{i_\alpha + m, i_\beta + m}^{[2]} = e_{\tau_{i^*}(i_\alpha), \tau_{i^*}(i_\beta)}^{[2]}$.

Case 4 $i^* + 1 + m \leq i_\alpha, i_\beta \leq 2m$
   Similar to case 3.

Case 5 $1 \leq i_\alpha < i^*, i^* + 1 \leq i_\beta \leq m$
   If $i_\alpha \neq 1$ or $i_\beta \neq m$, then since $H_m$ is a path and $i_\alpha < i^* \leq i_\beta - 1$, $(i_\alpha, i_\beta) \notin E^{[1]}$ or $E^{[2]}$. Now we consider the case where $i_\alpha = 1, i_\beta = m$. As $1 \leq i^* < m$, by the definition of $\tau$, there is $\tau_{i^*}(1) = 1$, and $\tau_{i^*}(m) = 2m$. Note that both $(1, m) \in E^{[1]}$ and $(1, 2m) \in E^{[2]}$ are true, and moreover, $e_{1,m}^{[1]} = e_{1,2m}^{[2]}$.

Case 6 $1 \leq i_\beta < i^*, i^* + 1 \leq i_\alpha \leq m$
   Similar to case 5.

Case 7 $1 + m \leq i_\alpha < i^* + m, i^* + 1 + m \leq i_\beta \leq 2m$
   Similar to case 5.

Case 8 $1 + m \leq i_\beta < i^* + m, i^* + 1 + m \leq i_\alpha \leq 2m$
   Similar to case 5.

Case 9 $1 \leq i_\alpha < i^*$ and $1 + m \leq i_\beta < i^* + m$
   Then $\tau_s(i_\alpha) = i_\alpha, \tau_s(i_\beta) = i_\beta$, and $(i_\alpha, i_\beta) \notin E^{[1]}$ or $E^{[2]}$.

Case 10 $1 \leq i_\beta < i^*$ and $1 + m \leq i_\alpha < i^* + m$
   Similar to case 9.

Case 11 $i^* + 1 \leq i_\alpha < m$ and $i^* + 1 + m \leq i_\beta \leq 2m$
   $(i_\alpha, i_\beta) \notin E^{[1]}$. $\tau_s(i_\alpha) = i_\alpha + m, \tau_s(i_\beta) = i_\beta - m$. Hence $(\tau_s(i_\alpha), \tau_s(i_\beta)) \notin E^{[2]}$ either.

Case 12 $i^* + 1 \leq i_\beta \leq m$ and $i^* + 1 + m \leq i_\alpha \leq 2m$
   Similar to case 11.

Case 13 $1 \leq i_\alpha < i^*$ and $i^* + 1 + m \leq i_\beta \leq 2m$
   $(i_\alpha, i_\beta) \notin E^{[1]}$ obviously. We also have $\tau_s(i_\alpha) = i_\alpha \in [1, i^*), \tau_s(i_\beta) = i_\beta - 1 \in [i^* + 1, m]$, and hence $(\tau_s(i_\alpha), \tau_s(i_\beta)) \notin E^{[2]}$.

Case 14 $1 \leq i_\beta < i^*$ and $i^* + 1 + m \leq i_\alpha \leq 2m$
   Similar to case 13.

Case 15 $1 + m \leq i_\alpha < i^* + m$ and $i^* + 1 \leq i_\beta \leq m$
   Similar to case 13.

Case 16 $1 + m \leq i_\beta < i^* + m$ and $i^* + 1 \leq i_\alpha \leq m$
Similar to case 13.

This concludes the proof of Lemma H.3.

□

Lemma H.3 completes the proof of the base case, and hence the induction argument for Lemma H.2.

□

$\forall s \in [2m]^k$, since $\eta(s) = \zeta_{\chi(s)}(s)$, and $\chi(s)$ satisfies $\forall j \in s$, either $\text{MOD}_m(j) < i$ or $\text{MOD}_m(j) \geq i + 2^T$, Lemma H.2 implies that at iteration $T$, we have $c_k^{(T)}(s) = c'^{(T)}_k(\zeta_{\chi(s)}(s)) = c'^{(T)}_k(\eta(s))$. Since we have shown that $\eta$ is a permutation on $[2m]^k$, this let's us conclude that

$$\{c_k^{(T)}(s) : s \in [2m]^k\} = \{c'^{(T)}_k(s) : s \in [2m]^k\}, \tag{28}$$

and therefore $k$-WL cannot distinguish between the two graphs in $T$ iterations. □

# I  Proof of Theorem 4.2 (2-IGNs are no more powerful than 2-WL)

Note that a 2-IGN takes as input a third-order tensor, $\boldsymbol{B}^{(0)}$, defined as in (1). If we use $\boldsymbol{B}^{(t)}$ to denote the output of the $t$th layer of the 2-IGN, then they are obtained iteratively by

$$\boldsymbol{B}^{(t+1)} = \sigma(L^{(t)}(\boldsymbol{B}^{(t)})) \tag{29}$$

*Proof.* For simplicity of notations, we assume $d_t = 1$ in every layer of a 2-IGN. The general case can be proved by adding more subscripts. For 2-WL, we use the definition in Appendix C except for omitting the subscript $k$ in $c_k^{(t)}$.

To start, it is straightforward to show (and we will prove it at the end) that the theorem can be deduced from the following lemma:

**Lemma I.1.** *Say $G^{[1]}$ and $G^{[2]}$ cannot be distinguished by the 2-WL. Then $\forall t \in \mathbb{N}$, it holds that*

$$\forall s, s' \in V^2, \text{ if } \boldsymbol{c}^{(t)}(s) = \boldsymbol{c}'^{(t)}(s'), \text{ then } \boldsymbol{B}_s^{(t)} = \boldsymbol{B}'^{(t)}_{s'} \tag{30}$$

This lemma can be shown by induction. To see this, first note that the lemma is equivalent to the statement that

$$\forall T \in \mathbb{N}, \forall t \leq T, (30) \text{ holds.}$$

This allows us to carry out an induction in $T \in \mathbb{N}$. For the base case $t = T = 0$, this is true because $\boldsymbol{c}^{(0)}$ and $\boldsymbol{c}'^{(0)}$ in WL and $\boldsymbol{B}^{(0)}$ and $\boldsymbol{B}'^{(0)}$ in 2-IGN are both initialized in the same way according to the subgraph isomorphism. To be precise, $\boldsymbol{c}^{(0)}(s) = \boldsymbol{c}'^{(t)}(s')$ if and only if the subgraph in $G^{[1]}$ induced by the pair of nodes $s$ is isomorphic to the subgraph in $G^{[2]}$ induced by the pair of nodes $s'$, which is also true if and only if $\boldsymbol{B}_s^{(0)} = \boldsymbol{B}'^{(0)}_{s'}$.

Next, to show that the induction step holds, we need to prove the following statement:

$$\forall T \in \mathbb{N}, \text{ if } \forall t \leq T - 1, (30) \text{ holds,}$$
$$\text{then } (30) \text{ also holds for } t = T.$$

To prove the consequent, we assume that for some $s, s' \in V^2$, there is $\boldsymbol{c}^{(T)}(s) = \boldsymbol{c}'^{(T)}(s')$, and then attempt to show that $\boldsymbol{B}_s^{(T)} = \boldsymbol{B}'^{(T)}_{s'}$. By the update rules of $k$-WL, the statement $\boldsymbol{c}^{(T)}(s) = \boldsymbol{c}'^{(T)}(s')$ implies that

$$\begin{cases} \boldsymbol{c}^{(T-1)}(s) = \boldsymbol{c}'^{(T-1)}(s') \\ \{\boldsymbol{c}^{(T-1)}(\tilde{s}) : \tilde{s} \in N_1(s)\} = \{\boldsymbol{c}'^{(T-1)}(\tilde{s}) : \tilde{s} \in N_1(s')\} \\ \{\boldsymbol{c}^{(T-1)}(\tilde{s}) : \tilde{s} \in N_2(s)\} = \{\boldsymbol{c}'^{(T-1)}(\tilde{s}) : \tilde{s} \in N_2(s')\} \end{cases} \tag{31}$$

**Case 1:** $s = (i, j) \in V^2$ **with** $i \neq j$

Let's first consider the case where $s = (i, j) \in V^2$ with $i \neq j$. In this case, we can also write $s' = (i', j') \in V^2$ with $i' \neq j'$, thanks to Lemma D.1. Then, note that $V^2$ can be written as the union of 9 disjoint sets that are defined depending on $s$:

$$V^2 = \bigcup_{w=1}^{9} A_{s,w},$$

where we define $A_{s,1} = \{(i,j)\}$, $A_{s,2} = \{(i,i)\}$, $A_{s,3} = \{(j,j)\}$, $A_{s,4} = \{(i,k) : k \neq i \text{ or } j\}$, $A_{s,5} = \{(k,i) : k \neq i \text{ or } j\}$, $A_{s,6} = \{(j,k) : k \neq i \text{ or } j\}$, $A_{s,7} = \{(k,j) : k \neq i \text{ or } j\}$, $A_{s,8} = \{(k,l) : k \neq l \text{ and } \{k,l\} \cap \{i,j\} = \emptyset\}$, and $A_{s,9} = \{(k,k) : k \notin \{i,j\}\}$. In this way, we partition $V^2$ into 9 different subsets, each of which consisting of pairs $(k,l)$ that yield a particular equivalence class of the 4-tuple $(i,j,k,l)$. Similarly, we can define $A_{s',w}$ for $w \in [9]$, which will also give us

$$V^2 = \bigcup_{w=1}^{9} A_{s',w}$$

Moreover, note that

$$N_1(s) = \bigcup_{w=1,3,7} A_{s,w}$$

$$N_2(s) = \bigcup_{w=1,2,4} A_{s,w}$$

$$N_1(s') = \bigcup_{w=1,3,7} A_{s',w}$$

$$N_2(s') = \bigcup_{w=1,2,4} A_{s',w}$$

Before proceeding, we make the following definition to simplify notations:

$$\mathfrak{C}_{s,w} = \{\boldsymbol{c}^{(T-1)}(\tilde{s}) : \tilde{s} \in A_{s,w}\}$$

$$\mathfrak{C}'_{s',w} = \{\boldsymbol{c}'^{(T-1)}(\tilde{s}) : \tilde{s} \in A_{s',w}\}$$

This allows us to rewrite (31) as

$$\mathfrak{C}_{s,1} = \mathfrak{C}'_{s',1} \tag{32}$$

$$\bigcup_{w=1,3,7} \mathfrak{C}_{s,w} = \bigcup_{w=1,3,7} \mathfrak{C}'_{s',w} \tag{33}$$

$$\bigcup_{w=1,2,4} \mathfrak{C}_{s,w} = \bigcup_{w=1,2,4} \mathfrak{C}'_{s',w} \tag{34}$$

Combining (32) and (33), we obtain

$$\bigcup_{w=3,7} \mathfrak{C}_{s,w} = \bigcup_{w=3,7} \mathfrak{C}'_{s',w} \tag{35}$$

Combining (32) and (34), we obtain

$$\bigcup_{w=2,4} \mathfrak{C}_{s,w} = \bigcup_{w=2,4} \mathfrak{C}'_{s',w} \tag{36}$$

Note that $V^2$ can also be partitioned into two disjoint subsets:

$$V^2 = \left( \bigcup_{w=1,4,5,6,7,8} A_{s,w} \right) \bigcap \left( \bigcup_{w=2,3,9} A_{s,w} \right),$$

where the first subset represent the edges: $\{(i,j) \in V^2 : i \neq j\}$ and the second subset represent the nodes: $\{(i,i) : i \in V\}$. Similarly,

$$V^2 = \left( \bigcup_{w=1,4,5,6,7,8} A_{s',w} \right) \bigcap \left( \bigcup_{w=2,3,9} A_{s',w} \right),$$

As shown in Lemma D.1, pairs of nodes that represent edges cannot share the same color with pairs of nodes the represent nodes in any iteration of 2-WL. Thus, we have

$$\left( \bigcup_{w=1,4,5,6,7,8} \mathfrak{C}_{s,w} \right) \bigcap \left( \bigcup_{w=2,3,9} \mathfrak{C}'_{s',w} \right) = \emptyset \tag{37}$$

$$\left( \bigcup_{w=1,4,5,6,7,8} \mathfrak{C}'_{s',w} \right) \bigcap \left( \bigcup_{w=2,3,9} \mathfrak{C}_{s,w} \right) = \emptyset \tag{38}$$

Combining (35) and (37) or (38), we get

$$\mathfrak{C}_{s,3} = \mathfrak{C}'_{s',3} \tag{39}$$

$$\mathfrak{C}_{s,7} = \mathfrak{C}'_{s',7} \tag{40}$$

Combining (36) and (37) or (38), we get

$$\mathfrak{C}_{s,2} = \mathfrak{C}'_{s',2} \tag{41}$$

$$\mathfrak{C}_{s,4} = \mathfrak{C}'_{s',4} \tag{42}$$

Thanks to symmetry between $(i,j)$ and $(j,i)$, as we work with undirected graphs, there is

$$\mathfrak{C}_{s,5} = \mathfrak{C}_{s,4} = \mathfrak{C}'_{s',4} = \mathfrak{C}'_{s',5} \tag{43}$$

$$\mathfrak{C}_{s,6} = \mathfrak{C}_{s,7} = \mathfrak{C}'_{s',7} = \mathfrak{C}'_{s',6} \tag{44}$$

In addition, since we assume that $G^{[1]}$ and $G^{[2]}$ cannot be distinguished by 2-WL, there has to be

$$\bigcup_{w=1}^{9} \mathfrak{C}_{s,w} = \bigcup_{w=1}^{9} \mathfrak{C}'_{s',w}$$

Combining this with (37) or (38), we get

$$\bigcup_{w=1,4,5,6,7,8} \mathfrak{C}_{s,w} = \bigcup_{w=1,4,5,6,7,8} \mathfrak{C}'_{s',w} \tag{45}$$

$$\bigcup_{w=2,3,9} \mathfrak{C}_{s,w} = \bigcup_{w=2,3,9} \mathfrak{C}'_{s',w} \tag{46}$$

Combining (45) with (32), (42), (43), (44), (40), we get

$$\mathfrak{C}_{s,8} = \mathfrak{C}'_{s',8} \tag{47}$$

Combining (46) with (41) and (39), we get

$$\mathfrak{C}_{s,9} = \mathfrak{C}'_{s',9} \tag{48}$$

Hence, in conclusion, we have that $\forall w \in [9]$,

$$\mathfrak{C}_{s,w} = \mathfrak{C}'_{s',w} \tag{49}$$

By the inductive hypothesis, this implies that $\forall w \in [9]$,

$$\{ \boldsymbol{B}_{\tilde{s}}^{(T-1)} : \tilde{s} \in A_{s,w} \} = \{ \boldsymbol{B}'^{(T-1)}_{\tilde{s}} : \tilde{s} \in A_{s',w} \} \tag{50}$$

Let us show how (50) may be leveraged. First, to prove that $\boldsymbol{B}_s^{(T)} = \boldsymbol{B}'^{(T)}_{s'}$, recall that

$$\boldsymbol{B}^{(T)} = \sigma(L^{(T)}(\boldsymbol{B}^{(T-1)}))$$
$$\boldsymbol{B}'^{(T)} = \sigma(L^{(T)}(\boldsymbol{B}'^{(T-1)})) \tag{51}$$

Therefore, it is sufficient to show that for all linear equivariant layer $L$, we have

$$L(\boldsymbol{B}^{(T-1)})_{i,j} = L(\boldsymbol{B}'^{(T-1)})_{i',j'} \tag{52}$$

Also, recall that

$$L(\boldsymbol{B}^{(T-1)})_{i,j} = \sum_{(k,l) \in V^2} T_{i,j,k,l} \boldsymbol{B}_{k,l} + Y_{i,j}$$

$$L(\boldsymbol{B}'^{(T-1)})_{i',j'} = \sum_{(k',l') \in V^2} T_{i',j',k',l'} \boldsymbol{B}'_{k',l'} + Y_{i',j'} \tag{53}$$

By the definition of the $A_{s,w}$'s and $A_{s',w}$'s, there is $\forall w \in [9], \forall (k,l) \in A_{s,w}, \forall (k',l') \in A_{s',w}$, we have the 4-tuples $(i,j,k,l) \sim (i',j',k',l')$, i.e., $\exists$ a permutation $\pi$ on $V$ such that $(i,j,k,l) = (\pi(i'), \pi(j'), \pi(k'), \pi(l'))$, which implies that $T_{i,j,k,l} = T_{i',j',k',l'}$. Therefore, together with (50), we have the following:

$$
\begin{aligned}
L(\boldsymbol{B}^{(T-1)})_{i,j} &= \sum_{(k,l) \in V^2} T_{i,j,k,l} \boldsymbol{B}_{k,l} + Y_{i,j} \\
&= \sum_{w=1}^{9} \sum_{(k,l) \in A_{s,w}} T_{i,j,k,l} \boldsymbol{B}_{k,l} + Y_{i,j} \\
&= \sum_{w=1}^{9} \sum_{(k',l') \in A_{s',w}} T_{i',j',k',l'} \boldsymbol{B}'_{k',l'} + Y_{i',j'} \\
&= L(\boldsymbol{B}'^{(T-1)})_{i',j'}
\end{aligned}
\tag{54}
$$

and hence $\boldsymbol{B}_{i,j}^{(T)} = \boldsymbol{B}'^{(T)}_{i'j'}$, which concludes the proof for the case that $s = (i,j)$ for $i \neq j$.

**Case 2:** $s = (i,i) \in V^2$
Next, consider the case $s = (i,i) \in V^2$. In this case, $s' = (i',i')$ for some $i' \in V$. This time, we write $V^2$ as the union of 5 disjoint sets that depend on $s$ (or $s'$):

$$
V^2 = \bigcup_{w=1}^{5} A_{s,w},
$$

where we define $A_{s,1} = \{(i,i)\}$, $A_{s,2} = \{(i,j) : j \neq i\}$, $A_{s,3} = \{(j,i) : j \neq i\}$, $A_{s,4} = \{(j,k) : j,k \neq i \text{ and } j \neq k\}$, and $A_{s,5} = \{(j,j) : j \neq i\}$. Similar for $s'$. We can also define $\mathfrak{C}_{s,w}$ and $\mathfrak{C}'_{s',w}$ as above. Note that

$$
\begin{aligned}
N_1(s) &= \bigcup_{w=1,3} A_{s,w} \\
N_2(s) &= \bigcup_{w=1,2} A_{s,w} \\
N_1(s') &= \bigcup_{w=1,3} A_{s',w} \\
N_2(s') &= \bigcup_{w=1,2} A_{s',w}
\end{aligned}
$$

Hence, we can rewrite (31) as

$$
\mathfrak{C}_{s,1} = \mathfrak{C}'_{s',1} \tag{55}
$$

$$
\bigcup_{w=1,3} \mathfrak{C}_{s,w} = \bigcup_{w=1,3} \mathfrak{C}'_{s',w} \tag{56}
$$

$$
\bigcup_{w=1,2} \mathfrak{C}_{s,w} = \bigcup_{w=1,2} \mathfrak{C}'_{s',w} \tag{57}
$$

Combining (55) with (56), we get

$$
\mathfrak{C}_{s,3} = \mathfrak{C}'_{s',3} \tag{58}
$$

Combining (55) with (57), we get

$$
\mathfrak{C}_{s,2} = \mathfrak{C}'_{s',2} \tag{59}
$$

Moreover, since we can decompose $V^2$ as

$$
\begin{aligned}
V^2 &= \Big( \bigcup_{w=1,5} A_{s,w} \Big) \bigcup \Big( \bigcup_{w=2,3,4} A_{s,w} \Big) \\
&= \Big( \bigcup_{w=1,5} A_{s',w} \Big) \bigcup \Big( \bigcup_{w=2,3,4} A_{s',w} \Big)
\end{aligned}
$$

with $\bigcup_{w=1,5} A_{s,w} = \bigcup_{w=1,5} A_{s',w}$ representing the nodes and $\bigcup_{w=2,3,4} A_{s,w} = \bigcup_{w=2,3,4} A_{s',w}$ representing the edges, we have

$$\left( \bigcup_{w=1,5} \mathfrak{C}_{s,w} \right) \cap \left( \bigcup_{w=2,3,4} \mathfrak{C}'_{s',w} \right) = \emptyset \tag{60}$$

$$\left( \bigcup_{w=1,5} \mathfrak{C}'_{s',w} \right) \cap \left( \bigcup_{w=2,3,4} \mathfrak{C}_{s,w} \right) = \emptyset \tag{61}$$

Since $G^{[1]}$ and $G^{[2]}$ cannot be distinguished by 2-WL, there is

$$\bigcup_{w=1}^{5} \mathfrak{C}_{s,w} = \bigcup_{w=1}^{5} \mathfrak{C}'_{s',w}$$

Therefore, combining this with (60) or (61), we obtain

$$\bigcup_{w=1,5} \mathfrak{C}_{s,w} = \bigcup_{w=1,5} \mathfrak{C}'_{s',w} \tag{62}$$

$$\bigcup_{w=2,3,4} \mathfrak{C}_{s,w} = \bigcup_{w=2,3,4} \mathfrak{C}'_{s',w} \tag{63}$$

Combining (62) with (55), we get

$$\mathfrak{C}_{s,5} = \mathfrak{C}'_{s',5} \tag{64}$$

Combining (63) with (59) and (58), we get

$$\mathfrak{C}_{s,4} = \mathfrak{C}'_{s',4} \tag{65}$$

Hence, in conclusion, we have that $\forall w \in [5]$,

$$\mathfrak{C}_{s,w} = \mathfrak{C}'_{s',w} \tag{66}$$

By the inductive hypothesis, this implies that $\forall w \in [5]$,

$$\{ \boldsymbol{B}_{\tilde{s}}^{(T-1)} : \tilde{s} \in A_{s,w} \} = \{ \boldsymbol{B}'^{(T-1)}_{\tilde{s}} : \tilde{s} \in A_{s',w} \} \tag{67}$$

Thus,

$$\begin{aligned}
L(\boldsymbol{B}^{(T-1)})_{i,i} &= \sum_{(k,l) \in V^2} T_{i,i,k,l} \boldsymbol{B}_{k,l} + Y_{i,i} \\
&= \sum_{w=1}^{5} \sum_{(k,l) \in A_{s,w}} T_{i,i,k,l} \boldsymbol{B}_{k,l} + Y_{i,i} \\
&= \sum_{w=1}^{5} \sum_{(k',l') \in A_{s',w}} T_{i',i',k',l'} \boldsymbol{B}'_{k',l'} + Y_{i',i'} \\
&= L(\boldsymbol{B}'^{(T-1)})_{i',i'}
\end{aligned}$$

and hence $\boldsymbol{B}_{i,j}^{(T)} = \boldsymbol{B}'^{(T)}_{i'j'}$, which concludes the proof for the case that $s = (i,i)$ for $i \in V$.

$\square$

Now, suppose we are given any 2-IGN with $T$ layers. Since $G^{[1]}$ and $G^{[2]}$ cannot be distinguished by 2-WL, together with Lemma D.1, there is

$$\{ \boldsymbol{c}^{(T)}((i,j)) : i,j \in V, i \neq j \} = \{ \boldsymbol{c}'^{(T)}((i',j')) : i',j' \in V, i' \neq j' \}$$

and

$$\{ \boldsymbol{c}^{(T)}((i,i)) : i \in V \} = \{ \boldsymbol{c}'^{(T)}((i',i')) : i' \in V \}$$

Hence, by the lemma, we have

$$\{ \boldsymbol{B}_{(i,j)}^{(T)} : i,j \in V, i \neq j \} = \{ \boldsymbol{B}'^{(T)}_{(i',j')} : i',j' \in V, i' \neq j' \}$$

and

$$\{\boldsymbol{B}_{(i,i)}^{(T)} : i \in V\} = \{\boldsymbol{B}'_{(i',i')}^{(T)} : i' \in V\}$$

Then, since the second-last layer $h$ in the 2-IGN can be written as

$$h(\boldsymbol{B}) = \alpha \sum_{i,j \in V, i \neq j} \boldsymbol{B}_{i,j} + \beta \sum_{i \in V} \boldsymbol{B}_{i,i} \tag{68}$$

there is

$$h(\boldsymbol{B}^{(T)}) = h(\boldsymbol{B}'^{(T)}) \tag{69}$$

and finally

$$m \circ h(\boldsymbol{B}^{(T)}) = m \circ h(\boldsymbol{B}'^{(T)}) \tag{70}$$

which means the 2-IGN yields identical outputs on the two graphs.

## J   Direct proof of Corollary 4.4 ($2$-IGNs are unable to induced-subgraph-count patterns of $3$ or more nodes)

*Proof.* The same counterexample as in the proof of Theorem 3.3 given in Appendix E applies here, as we are going to show below. Note that we only need to consider the non-clique case, since the set of counterexample graphs for the non-clique case is a superset of the set of counterexample graphs for the clique case.

Let $\boldsymbol{B}$ be the input tensor corresponding to $G^{[1]}$, and $\boldsymbol{B}'$ corresponding to $G^{[2]}$. For simplicity, we assume in the proof below that $d_0, ..., d_T = 1$. The general case can be proved in the same way but with more subscripts. (In particular, for our counterexamples, (74) can be shown to hold for each of the $d_0$ feature dimensions.)

Define a set $S = \{(1,2), (2,1), (1+m, 2+m), (2+m, 1+m), (1, 2+m), (2+m, 1), (1+m, 2), (2, 1+m)\}$, which represents the "special" edges that capture the difference between $G^{[1]}$ and $G^{[2]}$. We aim to show something like this:

$\forall t$,

$$\begin{cases} \boldsymbol{B}_{i,j}^{(t)} = \boldsymbol{B}'_{i,j}^{(t)}, \forall (i,j) \notin S \\ \boldsymbol{B}_{1,2}^{(t)} = \boldsymbol{B}'_{1+m,2}^{(t)}, \\ \boldsymbol{B}_{2,1}^{(t)} = \boldsymbol{B}'_{2,1+m}^{(t)}, \\ \boldsymbol{B}_{1+m,2+m}^{(t)} = \boldsymbol{B}'_{1,2+m}^{(t)} \\ \boldsymbol{B}_{2+m,1+m}^{(t)} = \boldsymbol{B}'_{2+m,1}^{(t)} \\ \boldsymbol{B}_{1,2+m}^{(t)} = \boldsymbol{B}'_{1+m,2+m}^{(t)}, \\ \boldsymbol{B}_{2+m,1}^{(t)} = \boldsymbol{B}'_{2+m,1+m}^{(t)}, \\ \boldsymbol{B}_{1+m,2}^{(t)} = \boldsymbol{B}'_{1,2}^{(t)} \\ \boldsymbol{B}_{2,1+m}^{(t)} = \boldsymbol{B}'_{2,1}^{(t)} \end{cases} \tag{71}$$

If this is true, then it is not hard to show that the 2-IGN returns identical outputs on $\boldsymbol{B}$ and $\boldsymbol{B}'$, which we will leave to the very end. To represent the different cases above compactly, we define a permutation $\eta_1$ on $V \times V$ in the following way. First, define the following permutations on $V$:

$$\kappa_1(i) = \begin{cases} \text{MOD}_{2m}(1+m), & \text{if } i \in \{1, 1+m\} \\ i, & \text{otherwise} \end{cases}$$

Next, define the permutation $\tau_1$ on $V \times V$:

$$\tau_1((i,j)) = (\kappa_1(i), \kappa_1(j))$$

and then $\eta_1$ as the restriction of $\tau_1$ on the set $S \subset V \times V$:

$$\eta_1((i,j)) = \begin{cases} \tau_1((i,j)), & \text{if } (i,j) \in S \\ (i,j), & \text{otherwise} \end{cases}$$

Thus, (71) can be rewritten as

$$\forall t, \boldsymbol{B}_{i,j}^{(t)} = \boldsymbol{B'}_{\eta_1((i,j))}^{(t)} \tag{72}$$

Before trying to prove (72), let's define $\kappa_2, \tau_2$ and $\eta_2$ analogously:

$$\kappa_2(i) = \begin{cases} \text{MOD}_{2m}(2+m), & \text{if } i \in \{2, 2+m\} \\ i, & \text{otherwise} \end{cases}$$

$$\tau_2((i,j)) = (\kappa_2(i), \kappa_2(j))$$

$$\eta_2((i,j)) = \begin{cases} \tau_2((i,j)), & \text{if } (i,j) \in S \\ (i,j), & \text{otherwise} \end{cases}$$

Thus, by symmetry, (72) is equivalent to

$$\forall t, \boldsymbol{B}_{i,j}^{(t)} = \boldsymbol{B'}_{\eta_1((i,j))}^{(t)} = \boldsymbol{B'}_{\eta_2((i,j))}^{(t)} \tag{73}$$

Because of the recursive relation (29), we will show (73) by induction on $t$. For the base case, it can be verified that

$$\boldsymbol{B}_{i,j}^{(0)} = \boldsymbol{B'}_{\eta_1((i,j))}^{(0)} = \boldsymbol{B'}_{\eta_2((i,j))}^{(0)} \tag{74}$$

thanks to the construction of $G^{[1]}$ and $G^{[2]}$. Moreover, if we define another permutation $V \times V$, $\zeta_1$:

$$\zeta_1((i,j)) = \begin{cases} (\text{MOD}_{2m}(i+m), \text{MOD}_{2m}(j+m)), & \\ \quad \text{if } j \in \{1, 1+m\}, i \notin \{2, 2+m\} & \\ \quad \text{or } i \in \{1, 1+m\}, j \notin \{2, 2+m\} & \\ (i,j), \text{ otherwise} & \end{cases} \tag{75}$$

then thanks to the symmetry between $(i,j)$ and $(i+m, j+m)$, there is

$$\boldsymbol{B}_{i,j}^{(0)} = \boldsymbol{B}_{\zeta_1((i,j))}^{(0)}, \; \boldsymbol{B'}_{i,j}^{(0)} = \boldsymbol{B'}_{\zeta_1((i,j))}^{(0)}$$

Thus, for the induction to hold, and since $\sigma$ applies entry-wise, it is sufficient to show that

**Lemma J.1.** *If*

$$\boldsymbol{B}_{i,j} = \boldsymbol{B}_{\zeta_1((i,j))}, \; \boldsymbol{B'}_{i,j} = \boldsymbol{B'}_{\zeta_1((i,j))} \tag{76}$$

$$\boldsymbol{B}_{i,j} = \boldsymbol{B'}_{\eta_1((i,j))} = \boldsymbol{B'}_{\eta_2((i,j))}, \tag{77}$$

*then*

$$L(\boldsymbol{B})_{i,j} = L(\boldsymbol{B})_{\zeta_1((i,j))}, \; L(\boldsymbol{B'})_{i,j} = L(\boldsymbol{B'})_{\zeta_1((i,j))} \tag{78}$$

$$L(\boldsymbol{B})_{i,j} = L(\boldsymbol{B'})_{\eta_1((i,j))} = L(\boldsymbol{B'})_{\eta_2((i,j))}, \tag{79}$$

*Proof of Lemma J.1:* Again, by symmetry between $(i,j)$ and $(i+m, j+m)$, (78) can be easily shown.

For (79), because of the symmetry between $\eta_1$ and $\eta_2$, we will only prove the first equality. By Maron et al. [40], we can express the linear equivariant layer $L$ by

$$L(\boldsymbol{B})_{i,j} = \sum_{(k,l)=(1,1)}^{(2m,2m)} T_{i,j,k,l} \boldsymbol{B}_{k,l} + Y_{i,j}$$

where crucially, $T_{i,j,k,l}$ depends only on the equivalence class of the 4-tuple $(i,j,k,l)$.

We consider eight different cases separately.

Case 1 $i, j \notin \{1, 2, 1+m, 2+m\}$

There is $\eta_1((i,j)) = (i,j)$, and $(i,j,k,l) \sim (i,j,\eta_1((k,l)))$, and thus $T_{i,j,k,l} = T_{i,j,\eta_1((k,l))}$. Therefore,

$$
\begin{aligned}
L(\boldsymbol{B}')_{\eta_1((i,j))} &= L(\boldsymbol{B}')_{i,j} \\
&= \sum_{(k,l)=(1,1)}^{(2m,2m)} T_{i,j,k,l}\boldsymbol{B}'_{k,l} + Y_{i,j} \\
&= \sum_{\eta_1((k,l))=(1,1)}^{(2m,2m)} T_{i,j,\eta_1((k,l))}\boldsymbol{B}'_{\eta_1((k,l))} + Y_{i,j} \\
&= \sum_{(k,l)=(1,1)}^{(2m,2m)} T_{i,j,\eta_1((k,l))}\boldsymbol{B}'_{\eta_1((k,l))} + Y_{i,j} \\
&= \sum_{(k,l)=(1,1)}^{(2m,2m)} T_{i,j,k,l}\boldsymbol{B}'_{\eta_1((k,l))} + Y_{i,j} \\
&= \sum_{(k,l)=(1,1)}^{(2m,2m)} T_{i,j,k,l}\boldsymbol{B}_{k,l} + Y_{i,j} \\
&= \boldsymbol{B}_{i,j}
\end{aligned}
$$

Case 2 $i \in \{1, 1+m\}, j \notin \{1, 2, 1+m, 2+m\}$

There is $\eta_1((i,j)) = (i,j)$, and $(i,j,k,l) \sim (i,j,\eta_2((k,l)))$, because $\eta_2$ only involves permutation between nodes 2 and $2+m$, while $i$ and $j \notin \{2, 2+m\}$. Thus, $T_{i,j,k,l} = T_{i,j,\eta_2((k,l))}$. Therefore,

$$
\begin{aligned}
L(\boldsymbol{B}')_{\eta_1((i,j))} &= L(\boldsymbol{B}')_{i,j} \\
&= \sum_{(k,l)=(1,1)}^{(2m,2m)} T_{i,j,k,l}\boldsymbol{B}'_{k,l} + Y_{i,j} \\
&= \sum_{\eta_2((k,l))=(1,1)}^{(2m,2m)} T_{i,j,\eta_2((k,l))}\boldsymbol{B}'_{\eta_2((k,l))} + Y_{i,j} \\
&= \sum_{(k,l)=(1,1)}^{(2m,2m)} T_{i,j,\eta_2((k,l))}\boldsymbol{B}'_{\eta_2((k,l))} + Y_{i,j} \\
&= \sum_{(k,l)=(1,1)}^{(2m,2m)} T_{i,j,k,l}\boldsymbol{B}'_{\eta_2((k,l))} + Y_{i,j} \\
&= \sum_{(k,l)=(1,1)}^{(2m,2m)} T_{i,j,k,l}\boldsymbol{B}_{k,l} + Y_{i,j} \\
&= \boldsymbol{B}_{i,j}
\end{aligned}
$$

Case 3 $j \in \{1, 1+m\}, i \notin \{1, 2, 1+m, 2+m\}$

Analogous to case 2.

Case 4 $i \in \{2, 2+m\}, j \notin \{1, 2, 1+m, 2+m\}$

There is $\eta_1((i,j)) = (i,j)$, and $(i,j,k,l) \sim (i,j,\eta_1((k,l)))$, because $\eta_1$ only involves permutation between nodes 1 and $1+m$, while $i$ and $j \notin \{1, 1+m\}$. Thus, $T_{i,j,k,l} = T_{i,j,\eta_1((k,l))}$. Therefore, we can apply the same proof as for case 2 here except for changing $\eta_2$'s to $\eta_1$'s.

**Case 5** $j \in \{2, 2+m\}$, $i \notin \{1, 2, 1+m, 2+m\}$

Analogous to case 4.

**Case 6** $(i, j) \in S$

Define one other permutation on $V \times V$, $\xi_1$, as $\xi_1((i,j)) =$

$$\begin{cases} (\text{MOD}_{2m}(i+m), j), & \text{if } \text{MOD}_m(j) = 1, \text{MOD}_m(i) \neq 1 \text{ or } 2 \\ (i, \text{MOD}_{2m}(j+m)), & \text{if } \text{MOD}_m(i) = 1, \text{MOD}_m(j) \neq 1 \text{ or } 2 \\ (i, j), & \text{otherwise} \end{cases}$$

It can be verified that

$$\xi_1 \circ \tau_1 = \eta_1 \circ \zeta_1$$

Moreover, it has the property that if $(i,j) \in S$, then

$$(i, j, k, l) \sim (i, j, \xi_1(k, l))$$

because $\xi_1$ only involves permutations among nodes not in $\{1, 2, 1+m, 2+m\}$ while $i, j \in \{1, 2, 1+m, 2+m\}$. Thus, we have

$$\begin{aligned}
(i, j, k, l) &\sim (\kappa_1(i), \kappa_1(j), \kappa_1(k), \kappa_1(l)) \\
&= (\tau_1(i,j), \tau_1(k,l)) \\
&= (\eta_1(i,j), \tau_1(k,l)) \\
&\sim (\eta_1(i,j), \xi_1 \circ \tau_1(k,l)) \\
&= (\eta_1(i,j), \eta_1 \circ \zeta_1(k,l)),
\end{aligned}$$

implying that $T_{i,j,k,l} = T_{\eta_1(i,j),\eta_1 \circ \zeta_1(k,l)}$. In addition, as $\eta_1((i,j)) \sim (i,j)$, there is $Y_{\eta_1((i,j))} = Y_{i,j}$. Moreover, by (76),

$$\boldsymbol{B}'_{\eta_1 \circ \zeta_1((k,l))} = \boldsymbol{B}'_{\eta_1((k,l))} = \boldsymbol{B}_{k,l}$$

Therefore,

$$\begin{aligned}
L(\boldsymbol{B}')_{\eta_1((i,j))} &= \sum_{(k,l)=(1,1)}^{(2m,2m)} T_{\eta((i,j)),k,l} \boldsymbol{B}'_{k,l} + Y_{\eta_1((i,j))} \\
&= \sum_{\eta_1 \circ \zeta_1((k,l))=(1,1)}^{(2m,2m)} T_{\eta_1((i,j)),\eta_1 \circ \zeta_1((k,l))} \boldsymbol{B}'_{\eta_1 \circ \zeta_1((k,l))} + Y_{\eta_1((i,j))} \\
&= \sum_{(k,l)=(1,1)}^{(2m,2m)} T_{\eta_1((i,j)),\eta_1 \circ \zeta_1((k,l))} \boldsymbol{B}'_{\eta_1 \circ \zeta_1((k,l))} + Y_{\eta_1((i,j))} \\
&= \sum_{(k,l)=(1,1)}^{(2m,2m)} T_{i,j,k,l} \boldsymbol{B}_{k,l} + Y_{i,j} \\
&= \boldsymbol{B}_{i,j}
\end{aligned}$$

**Case 7** $i, j \in \{1, 1+m\}$

There is $\eta_1(i,j) = (i,j)$ and $(i,j,k,l) \sim (i,j,\eta_2((k,l)))$. Thus, $T_{i,j,k,l} = T_{i,j,\eta_2((k,l))}$, and the rest of the proof proceeds as for case 2.

**Case 8** $i, j \notin \{1, 1+m\}$

There is $\eta_1(i,j) = (i,j)$ and $(i,j,k,l) \sim (i,j,\eta_1((k,l)))$. Thus, $T_{i,j,k,l} = T_{i,j,\eta_1((k,l))}$, and the rest of the proof proceeds as for case 4.

$\square$

With the lemma above, (72) can be shown by induction as a consequence. Thus,

$$\boldsymbol{B}_{i,j}^{(T)} = \boldsymbol{B}_{\eta_1(i,j)}^{(T)}$$

Maron et al. [40] show that the space of linear invariant functions on $\mathbb{R}^{n \times n}$ is two-dimensional, and so for example, the second-last layer $h$ in the 2-IGN can be written as

$$h(\boldsymbol{B}) = \alpha \sum_{i,j=(1,1)}^{(2m,2m)} \boldsymbol{B}_{i,j} + \beta \sum_{i=1}^{2m} \boldsymbol{B}_{i,i}$$

for some $\alpha, \beta \in \mathbb{R}$. Then since $\eta_1$ is a permutation on $V \times V$ and also is the identity map when restricted to $\{(i,i) : i \in V\}$, we have

$$
\begin{aligned}
h(\boldsymbol{B'}^{(T)}) =& \alpha \sum_{(i,j)=(1,1)}^{(2m,2m)} \boldsymbol{B'}_{i,j}^{(T)} + \beta \sum_{i=1}^{2m} \boldsymbol{B'}_{i,i}^{(T)} \\
=& \alpha \sum_{(i,j)=(1,1)}^{(2m,2m)} \boldsymbol{B'}_{\eta_1((i,j))}^{(T)} + \beta \sum_{i=1}^{2m} \boldsymbol{B'}_{\eta_1((i,i))}^{(T)} \\
=& \alpha \sum_{(i,j)=(1,1)}^{(2m,2m)} \boldsymbol{B}_{i,j}^{(T)} + \beta \sum_{i=1}^{2m} \boldsymbol{B}_{i,i}^{(T)} \\
=& h(\boldsymbol{B}^{(T)})
\end{aligned}
$$

Therefore, finally,

$$m \circ h(\boldsymbol{B}^{(T)}) = m \circ h(\boldsymbol{B'}^{(T)})$$

$\square$

## K  Leveraging sparse tensor operations for LRP

Following our definition of Deep LRP in (4), in each layer, for each egonet $G_{i,l}^{[\text{ego}]}$ and each ordered subset $\tilde{\pi} \in \tilde{S}_{i,l}^{k\text{-BFS}}$ of nodes in $G_{i,l}^{[\text{ego}]}$, we need to compute the tensor $\tilde{\pi} \star \boldsymbol{B}_{i,l}^{[\text{ego}]}(H^{(t-1)})$ out of the hidden node states of the previous layer, $H^{(t-1)}$. This is compuationally challenging for stacking multiple layers. Moreover, the tensor operations involved in (4) are dense. In particular, if we batch multiple graphs together, the computational complexity grows quadratically in the number of graphs in a batch, whereas a more reasonable cost would be linear with respect to batch size. In this section, we outline an approach to improve efficiency in implementation via pre-computation and sparse tensor operations. Specifically, we propose to represent the mapping from an $H$ to the set of all $\tilde{\pi} \star \boldsymbol{B}_{i,l}^{[\text{ego}]}(H)$'s as a sparse matrix, which can be pre-computed and then applied in every layer. We will also define a similar procedure for the edge features.

The first step is to translate the *local* definitions of $\tilde{\pi} \star \boldsymbol{B}_{i,l}^{[\text{ego}]}$ in (4) to a *global* definition. The difference lies in the fact that $\boldsymbol{B}_{i,l}^{[\text{ego}]}$ implicitly defines a local node index for each node in the egonet, $G_{i,l}^{[\text{ego}]}$ – e.g., $(\boldsymbol{B}_{i,l}^{[\text{ego}]})_{j,j,:}$ gives the node feature of the $j$th node in $G_{i,l}^{[\text{ego}]}$ according to this local index, which is not necessarily the $j$th node in the whole graph, $G$. To deal with this notational subtlety, for each ordered subset $\tilde{\pi} \in \tilde{S}_{i,l}^{k\text{-BFS}}$, we associate with it an ordered subset $\Pi[\tilde{\pi}]$ with elements in $V$, such that the $(\tilde{\pi}(j))$th node in $G_{i,l}^{[\text{ego}]}$ according to the local index is indexed to be the $(\Pi[\tilde{\pi}](j))$th node in the whole graph. Thus, by this definition, we have $\Pi[\tilde{\pi}] \star \boldsymbol{B} = \tilde{\pi} \star \boldsymbol{B}_{i,l}^{[\text{ego}]}$.

Our next task is to efficiently implement the mapping from an $H$ to each $\Pi[\tilde{\pi}] \star \boldsymbol{B}(H)$. We propose to represent this mapping as a sparse matrix. To illustrate, below we consider the example of Deep LRP-$l$-$k$ with $l = 1$, and Figure 5 illustrates each step in a layer of Deep LRP-1-3 in particular. For Deep LRP-1-$k$, each ordered subset $\tilde{\pi} \in \tilde{S}_{i,1}^{k\text{-BFS}}$ consists of $(k+1)$ nodes, and therefore the first two dimensions of $\Pi[\tilde{\pi}] \star \boldsymbol{B} = \tilde{\pi} \star \boldsymbol{B}_{i,l}^{[\text{ego}]}$ are $(k+1) \times (k+1)$. We use the following definition of $\boldsymbol{B}$, which is slightly simpler than (1) by neglecting the adjacency matrix (whose information is already contained in the edge features): $\boldsymbol{B} \in \mathbb{R}^{n \times n \times d}$, with $d = \max(d_n, d_e)$, and

$$
\begin{aligned}
\mathbf{B}_{i,i,1:d_n} &= x_i , \quad \forall i \in V = [n] , \\
\mathbf{B}_{i,j,1:d_e} &= e_{i,j} , \quad \forall (i,j) \in E .
\end{aligned}
\tag{80}
$$

Similarly, for $H \in \mathbb{R}^{n \times d'}$, $\boldsymbol{B}(H)$ is defined to be an element of $\mathbb{R}^{n \times n \times \max(d', d_e)}$, with

$$
\begin{aligned}
\mathbf{B}_{i,i,1:d'} &= H_i , \quad \forall i \in V = [n] , \\
\mathbf{B}_{i,j,1:d_e} &= e_{i,j} , \quad \forall (i,j) \in E .
\end{aligned}
\tag{81}
$$

Below, we assume for the simplicity of presentation that $d_n = d_e = d'$. We let $|E|$ denote the number of edges in $G$. Define $Y \in \mathbb{R}^{|E| \times d_e}$ to be the matrix of edge features, where $Y_q$ is the feature vector of the $q$th edge in the graph according to some ordering of the edges. Let $P_i$ be the cardinality of $\tilde{S}_{i,l}^{k\text{-BFS}}$, and define $P = \sum_{i \in [n]} P_i$, where the summation is over all nodes in the graph. Note that these definitions can be generalized to the case where we have a batch of graphs.

We define *Node_to_perm*, denoted by $\mathbf{N2P}$, which is a matrix of size $((k+1)^2 P) \times N$ with each entry being 0 or 1. The first dimension corresponds to the flattening of the first two dimension of $\Pi[\tilde{\pi}] \star \boldsymbol{B}$ for all legitimate choices of $\tilde{\pi}$. Hence, each row corresponds to one of the $(k+1) \times (k+1)$ "slots" of the first two dimension of $\Pi[\tilde{\pi}] \star \boldsymbol{B}$ for some $\tilde{\pi}$. In addition, each column of $\mathbf{N2P}$ corresponds to a node in $G$. Thus, each entry $(m, j)$ of $\mathbf{N2P}$ is 1 if and only if the "slot" indexed by $m$ is filled by the $H_j$. By the definition of $\boldsymbol{B}(H)$, $\mathbf{N2P}$ is a sparse matrix. For the edge features, we similarly define *Edge_to_perm*, denoted by $\mathbf{E2P}$, with size $((k+1)^2 P) \times |E|$. Similar to $\mathbf{N2P}$, each entry $(m, q)$ of $\mathbf{E2P}$ is 1 if and only if the "slot" indexed by $m$ is filled by the $e_{j_1, j_2}$, where $(j_1, j_2)$ is the $q$th edge.

Hence, by these definitions, the list of the vectorizations of all $\tilde{\pi} \star \boldsymbol{B}_{i,l}^{[\text{ego}]}(H)$ can be obtained by

$$
\text{RESHAPE}\left(\mathbf{N2P} \cdot H + \mathbf{E2P} \cdot Y\right) \in \mathbb{R}^{P \times ((k+1)^2 d)} ,
\tag{82}
$$

where RESHAPE is a tensor-reshapping operation that splits the first dimension from $(k+1)^2 P$ to $P \times (k+1)^2$. Hence, with our choice of $f$ to be an MLP on the vectorization of its tensorial input, the list of all $f(\tilde{\pi} \star \boldsymbol{B}_{i,l}^{[\text{ego}]}(H))$ is obtained by

$$
\mathbf{MLP1}\left(\text{RESHAPE}\left(\mathbf{N2P} \cdot H + \mathbf{E2P} \cdot Y\right)\right) ,
\tag{83}
$$

where $\mathbf{MLP1}$ acts on the second dimension.

Next, we define the $\alpha_{\tilde{\pi}}$ factor as the output of an MLP applied to the relevant node degrees. When $l = 1$, we implement it as $\mathbf{MLP2}(D_i)$, the output of an MLP applied to the degree of the root node $i$ of the egonet. When $k = 1$, since each $\tilde{\pi} \in \tilde{S}_{i,l}^{k\text{-BFS}}$ consists of nodes on a path of length at most $(l+1)$ starting from node $i$, we let $\alpha_{\tilde{\pi}}$ be the output of an MLP applied to the concatenation of the degrees of all nodes on the path, as discussed in the main text. This step is also compatible with sparse operations similar to (83), in which we substitute $H$ with the degree vector $D \in \mathbb{R}^{n \times 1}$ and neglect $Y$. In the $l = 1$ case, the list of all the list of all $\alpha_{\tilde{\pi}} f(\tilde{\pi} \star \boldsymbol{B}_{i,l}^{[\text{ego}]}(H))$ is obtained by

$$
\mathbf{MLP1}\left(\text{RESHAPE}\left(\mathbf{N2P} \cdot H + \mathbf{E2P} \cdot Y\right)\right) \odot \mathbf{MLP2}(D_i)
\tag{84}
$$

Note that the output dimensions of $\mathbf{MLP1}$ and $\mathbf{MLP2}$ are chosen to be the same, and $\odot$ denotes the element-wise product between vectors.

The final step is to define *Permutation_pooling*, denoted by $\mathbf{PPL}$, which is a sparse matrix in $\mathbb{R}^{N \times P}$. Each non-zero entry at position $(j, p)$ means that the $p$-th $\tilde{\pi}$ among all $P$ of them for the whole graph (or a batch of graphs) contributes to the representation of node $i$ in the next layer. In particular, sum-pooling corresponds to setting all non-zero entries in $\mathbf{PPL}$ as 1, while average-pooling corresponds to first setting all non-zero entries in $\mathbf{PPL}$ as 1 and then normalizing it for every row, which is equivalent to having the factor $\frac{1}{|\tilde{S}_{i,1}^{k\text{-BFS}}|}$ in (4).

Therefore, we can now write the update rule (4) for LRP-1-$k$ as

$$
H_i^{(t)} = \mathbf{PPL} \cdot \left[\mathbf{MLP1}^{(t)}\left(\text{RESHAPE}\left(\mathbf{N2P} \cdot H^{(t-1)} + \mathbf{E2P} \cdot Y\right)\right) \odot \mathbf{MLP2}^{(t)}(D_i)\right],
\tag{85}
$$

## L  Theoretical limitations of GraphSAGE in substructure counting

In order for substructure counting to be well-defined, we do not consider random node sampling and only consider GraphSAGE with aggregation over a full neighborhood. If only 1-hop neighborhood is

$\mathbb{R}^{N \times d^{(t)}}$  Node Rep $\{h_i^{(t)}\}$  Edge Rep $\{e_{i,j}^{(t)}\}$  $\mathbb{R}^{|E| \times d^{(t)}}$

Node_to_perm   Edge_to_perm

$\mathbb{R}^{(16 \cdot P) \times d^{(t)}}$  Permuation Rep

RESHAPE

$\mathbb{R}^{P \times (16 \cdot d^{(t)})}$  Permuation Rep

MLP1$^{(t)}$

$\mathbb{R}^{P \times d^{(t+1)}}$  Permuation Rep   Node degree $\{D_i\}$

Permutation_Pooling   MLP2$^{(t)}$

$\mathbb{R}^{N \times d^{(t+1)}}$  Node Rep $\{h_i^{(t+1)}\}$

Figure 5: Illustration of the $t$-th local relational pooling layer in Deep LRP-1-3. Rounded rectangles denote representations (Rep) after each operation (denoted as arrows).

Figure 6: A pair of non-isomorphic attributed graphs that GraphSAGE cannot distinguish.

used for aggregation in each iteration, its expressive power is upper-bounded by that of WL, just like MPNNs. If multi-hop neighborhood is used for aggregation, the question becomes more interesting. Compared to LRP, however, GraphSAGE aggregates neighborhood information as a set or sequence rather than a tensor, which results in a loss of the information of the edge features and high-order structures. In particular,

1. The original GraphSAGE does not consider edge features. Even if we allow it to incorporation edge feature information via augmenting the node features by applying an invariant function to the features of its immediate edges (e.g. summing or averaging), GraphSAGE cannot distinguish the pairs of graphs shown in Figure 6, for example, while LRP-1-2 can.

2. GraphSAGE cannot distinguish the pair of 12-circular graphs $C_{12}(1,3)$ and $C_{12}(1,5)$ (see [61]), no matter the hop-size being used, because $\forall k$, the $k$-hop neighborhood of every node

in the two graphs has the same size. This means GraphSAGE cannot count the number of 4-cycles as either *subgraphs* or *induced subgraphs*, whereas LRP-2-2 is able to.

Further, Table 1 shows the performance on the synthetic tasks of GraphSAGE + LSTM using full 1-hop neighborhood for aggregation. We see that it can count stars but not triangles, consistent with the limitation of the information in the 1-hop neighborhood, in the same way as MPNNs.

# M  Additional details of the numerical experiments

## M.1  Models

As reported in Section 6, we run experiments on synthetic and real datasets using different GNN models. Below are some details regarding their architecture and implementation:

- **LRP-$l$-$k$**: Local Relational Pooling with egonet depth $l$ and $k$-truncated BFS, as described in the main text. For LRP-1-3, for example, with $d$ being the dimension of the initial tensor representation, $\boldsymbol{B}$, we define

$$\tilde{f}_{\text{LRP}}^{1,3}(G) = \mathbf{W}_1 \sum_{i \in V} \sigma \left[ \frac{\text{MLP}(D_i)}{|\tilde{S}_{i,1}^{3\text{-BFS}}|} \odot \sum_{\tilde{\pi} \in \tilde{S}_{i,1}^{3\text{-BFS}}} f_*(\tilde{\pi} \circ \boldsymbol{B}_{i,1}^{[\text{ego}]}) \right], \tag{86}$$

  where $D_i$ is the degree of node $i$, $\sigma$ is ReLU, MLP maps from $\mathbb{R}$ to $\mathbb{R}^H$, where $H$ is the hidden dimension, $\mathbf{W}_1 \in \mathbb{R}^{1 \times H}$ and $\forall p \in [H]$, $(f_*(\mathbf{X}))_p = \tanh(\sum \mathbf{W}_{2,p} \odot \mathbf{X}) \in \mathbb{R}$ with $\mathbf{W}_{2,p} \in \mathbb{R}^{4 \times 4 \times d}$. Note that each $\tilde{\pi} \in \tilde{S}_{i,1}^{3\text{-BFS}}$ is an ordered set of 4 nodes that begin with node $i$, and $\tilde{\pi} \circ \boldsymbol{B}_{i,1}^{[\text{ego}]}$ is a $4 \times 4 \times d$ tensor such that $(\tilde{\pi} \circ \boldsymbol{B}_{i,1}^{[\text{ego}]})_{j,j,:} = (\boldsymbol{B}_{i,1}^{[\text{ego}]})_{\tilde{\pi}(j),\tilde{\pi}(j),:}$. As discussed in the main text, $\text{MLP}(D_i)$ plays the role of $\alpha_{\tilde{\pi}}$, which adaptively learns an invariant function over permutation, such as summing and averaging.

  The nonlinear activation functions are chosen between ReLU and tanh by hand. The models are trained using the Adam optimizer [30] with learning rate 0.1. The number of hidden dimensions is searched in $\{1, 8, 16, 64, 128\}$.

- **Deep LRP-$l$-$k$**: The nonlinear activation functions are ReLU. For synthetic experiments, we set the depth of the model as 1. The number of hidden dimensions is searched in $\{64, 128\}$. We use summation for the final graph-level aggregation function. For real experiments, we search the depth of the model in $\{4, 5, 6, 7, 8, 10, 12, 20, 24\}$. The number of hidden dimensions is searched in $\{8, 16, 32, 50, 100, 128, 150, 200, 256, 300, 512\}$. The final graph-level aggregation function is average. We involve Batch Normalization [22] and Jumping Knowledge [65]. On ogbg-molhiv, we utilize AtomEncoder and BondEncoder following the official implementation of GIN [64] on the OGB leaderboard [20]. The models are trained using the Adam optimizer [30] with learning rate searched in $\{0.01, 0.005, 0.001, 0.0001\}$.

- **2-IGN**: The 2nd-order Invariant Graph Networks proposed by Maron et al. [40]. In our synthetic experiments, we chose 8 hidden dimensions for the invariant layers and 16 hidden dimensions for the output MLP. The models are trained using the Adam optimizer with learning rate 0.1. The numbers of hidden dimensions are searched in $\{(16, 32), (8, 16), (64, 64)\}$.

- **PPGN**: The Provably Powerful Graph Network model proposed in Maron et al. [39]. In our synthetic experiments, we choose the depth of the model to be 4 and select the hidden dimension in $\{16, 64\}$. The models are trained using the Adam optimizer [30] with learning rate searched in $\{0.01, 0.001, 0.0001, 0.00001\}$. The depth of each MLP involved in the model is 2.

- **GCN**: The Graph Convolutional Network proposed by Kipf and Welling [31]. In our experiments, we adopt a 4-layer GCN with 128 hidden dimensions. The models are trained using the Adam optimizer with learning rate 0.01. The number of hidden dimensions is searched in $\{8, 32, 128\}$. The depth is searched in $\{2, 3, 4, 5\}$.

- **GIN**: The Graph Isomorphism Network proposed by Xu et al. [64]. In our experiments, we adopt a 4-layer GIN with 32 hidden dimensions. The models are trained using the Adam optimizer with learning rate 0.01. The number of hidden dimensions is searched in $\{8, 16, 32, 128\}$.

- **sGNN**: Spectral GNN with operators from family $\{\mathbf{I}, \mathbf{A}, \min(\mathbf{A}^2, 1)\}$. In our experiments, we adopt a 4-layer sGNN with 128 hidden dimensions. The models are trained using the Adam optimizer with learning rate 0.01. The number of hidden dimensions is searched in $\{8, 128\}$.
- **GraphSAGE**: GraphSAGE [18] using LSTM [19] for aggregation over the full 1-hop neighborhood. In our experiments, we adopt a 5-layer GraphSAGE with 16 hidden dimensions. The models are trained using the Adam optimizer with learning rate 0.1.

For the experiments on substructure counting in random graphs, for GCN, GIN and sGNN, we always train four variants for each architecture, depending on whether Jump Knowledge [65] or Batch Normalization [22] is included or not. All models are trained for 100 epochs. Learning rates are searched in $\{1, 0.1, 0.05, 0.01\}$. We pick the best model with the lowest MSE loss on validation set to generate results.

## M.2 Counting substructures in random graphs

### M.2.1 Dataset generation

We generate two synthetic datasets of random unattributed graphs. The first one is a set of 5000 Erdős-Renyi random graphs denoted as $ER(m, p)$, where $m = 10$ is the number of nodes in each graph and $p = 0.3$ is the probability that an edge exists. The second one is a set of 5000 random regular graphs [57] denoted as $RG(m, d)$, where $m$ is the number of nodes in each graph and $d$ is the node degree. We uniformly sample $(m, d)$ from $\{(10, 6), (15, 6), (20, 5), (30, 5)\}$. We also randomly delete $m$ edges in each graph from the second dataset. For both datasets, we randomly split them into training-validation-test sets with percentages 30%-20%-50%. For the attributed task, we mark nodes with even indices as red and nodes with odd indices as blue, and set the color as node feature using 1-hot encoding.

### M.2.2 Additional results

For the synthetic experiments, we design five substructure-counting tasks with patterns illustrated in Figure 3. In Section 6, we show the results for the subgraph-count of 3-stars and the induced-subgraph-count of triangles. In this section, we give results for the the remaining patterns: tailed triangles, chordal cycles and attributed triangles. As we see in Table M.2.2, while Deep LRP-1-3 achieves the best overall performance, all three models perform well in learning the induced-subgraph-count of each of these three patterns on at least one of the two synthetic datasets.

Table 5: Performance of the different models on learning the induced-subgraph-count of tailed triangles, chordal cycles and attributed triangles on the two datasets, measured by test MSE divided by variance of the ground truth counts (given in Table 6). Shown here are the best and the median performances of each model over five runs.

| | Erdős-Renyi | | | | Random Regular | | | |
|---|---|---|---|---|---|---|---|---|
| | Tailed Triangle | | Chordal Cycle | | Tailed Triangle | | Chordal Cycle | |
| | top 1 | top 3 | top 1 | top 3 | top 1 | top 3 | top 1 | top 3 |
| LRP-1-3 | 7.61E-5 | 1.94E-4 | 5.97E-4 | 7.73E-4 | 9.80E-5 | 2.01E-4 | 8.19E-5 | 1.63E-4 |
| Deep LRP-1-3 | 3.96E-5 | 1.35E-4 | 6.50E-5 | 8.96E-5 | 1.60E-5 | 2.02E-4 | 3.83E-9 | 3.99E-6 |
| PPGN | 7.11E-3 | 2.03E-2 | 2.14E-2 | 1.31E-1 | 2.29E-3 | 6.88E-3 | 5.90E-4 | 3.12E-2 |

| | Erdős-Renyi | | | | Random Regular | | | |
|---|---|---|---|---|---|---|---|---|
| | Attributed Triangle | | | | Attributed Triangle | | | |
| | top 1 | top 3 | | | top 1 | top 3 | | |
| LRP-1-3 | 9.23E-4 | 2.12E-3 | | | 4.50E-1* | 4.72E-1* | | |
| Deep LRP-1-3 | 1.48E-4 | 1.35E-3 | | | 9.06E-5 | 5.05E-4 | | |
| PPGN | 2.58E-5 | 8.02E-5 | | | 4.30E-1* | 4.33E-1* | | |

## M.3 Molecular prediction tasks

### M.3.1 ogbg-molhiv

The molecular dataset ogbg-molhiv from the Open Graph Benchmark (OGB) contains 41127 graphs, with 25.5 nodes and 27.5 edges per graph on average, and the task is to predict 1 target graph-level

Table 6: Variance of the ground truth labels for each synthetic task.

| Task | Erdős-Renyi | Random Regular |
|---|---|---|
| 3-star | 311.2 | 316.0 |
| triangle | 7.353 | 9.437 |
| tailed triangle | 67.53 | 163.7 |
| chordal cycle | 9.609 | 11.40 |
| attributed triangle | 2.111 | 2.709 |

label. Each graph represents a molecule, where the nodes represent atoms and the edges represent chemical bonds. We use binary cross entropy as the loss function, and we utilize the official APIs including an evaluator provided by OGB (version 1.1.1) [20].

ogbg-molhiv adopts the *scaffold splitting* procedure that splits the data based on their two-dimensional structural frameworks. Because of this, more training epochs might lead to overfitting and therefore worse performance on the test set. Hence, we report the results of LRP-1-3 trained different number of epochs: "LRP-1-3" is trained for 100 epochs, same as other models reported on the OGB leaderboard, and "LRP-1-3 (ES)" is trained for 20 epochs only. To ensure the reproducibility of our results, LRP-1-3 (ES) is run with 35 random seeds, from 0 to 34.

We report the average training time of Deep LRP-1-3 on ogbg-molhiv in Table 7. We can see that Deep LRP-1-3 approximately takes 5-8× time as much as GIN. However, the ratio goes down to 3-5× when we utilize more numbers of workers to load the data, because the dataloader involves batching operations as defined in Appendix K. We also split the training time for one epoch into several components in Table 8. It turns out the operations **N2P**, **E2P** and **Ppl** account for most of the forward running time, which indicates a possible direction to optimize the current implementation.

Table 7: Training time per epoch for different GNNs on ogbg-molhiv with batch size of 64. All results are generated from a computing node with a GTX 1080Ti, 4 CPUs and 32GB RAM. "#workers" stands for the number of workers in Dataloader of PyTorch. "Ours-2" is the model reported as "Deep LRP-1-3" in Table 2 while "Ours-3" is the model reported as "Deep LRP-1-3 (ES)".

| model | time/epoch (sec) | #params | #workers |
|---|---|---|---|
| GIN | 26 | 189K | 0 |
| GIN | 26 | 189K | 4 |
| Ours-1 | 133 | 166K | 0 |
| Ours-1 | 82 | 166K | 4 |
| Ours-2 | 136 | 98K | 0 |
| Ours-2 | 83 | 98K | 4 |
| Ours-3 | 194 | 630K | 0 |
| Ours-3 | 122 | 630K | 4 |

Table 8: Components of training time in an epoch. The setting is the same as that in Table 7.

| model | #workers | total time | forward | N2P&E2P | Ppl | backward |
|---|---|---|---|---|---|---|
| Ours-1 | 4 | 81.8 | 39.3 | 16.4 | 17.2 | 13.2 |

### M.3.2 QM9

QM9 has 134K graphs and 12 graph-level target labels for regression. The data is randomly split into 80% for training, 10% for validation and 10% for testing. For training loss, we use the 12-target average of the normalized Mean Absolute Error, where normalization means dividing by the standard deviation of all training labels in the dataset for each of the 12 targets. We report this averaged normalized MAE as "Loss" in the last row Table 4.

### M.3.3 ZINC

ZINC [23] is a real-world molecular dataset of 250K graphs. We follow the setting of [14] that selects 12K graphs for regression out of the entire dataset. The dataset is split into 10K/1K/1K for training/validation/testing. We use Mean Absolute Error as the loss for training, validation and testing. Baselines in Table 3 are picked as the best results from [14] regardless of numbers of parameters. Here we also list results with numbers of parameters in Table 9. It turns out our models outperforms all other baselines with the same level of numbers of parameters.

Following [14], we train Deep LRPs with a learning rate scheduler, in which the learning rate decay factor is 0.5 and the patience value for validation loss is 10. The stopping criterion is whether the current learning rate is smaller than 1 percent of the initial learning rate.

Table 9: Additional ZINC test results measured by Mean Abosolute Error (MAE). All baselines are taken from [14, 52]. $^{\dagger}$: Also reported in Table 3.

| Model | #Params | Testing MAE | #Params | Testing MAE |
|---|---|---|---|---|
| MLP | 106970 | 0.681±0.005 | 2289351 | 0.704 ±0.003 |
| GCN | 103077 | 0.469±0.002 | 2189531 | 0.479±0.007 |
| GraphSAGE | 105031 | 0.410±0.005 | 2176751 | 0.439±0.006 |
| GIN | 103079 | 0.408±0.008 | 2028508 | 0.382±0.008 |
| DiffPool | 110561 | 0.466±0.006 | 2291521 | 0.448±0.005 |
| GAT | 102385 | 0.463±0.002 | 2080881 | 0.471±0.005 |
| MoNet | 106002 | 0.407±0.007 | 2244343 | 0.372±0.01 |
| GatedGCN | 105875 | 0.363±0.009 | 2134081 | 0.338±0.003 |
| LRGA + GatedGCN | 94457 | 0.367±0.008 | 1989730 | 0.285±0.01 |
| Deep LRP-7-1 | 92073 | **0.317±0.031** | 1695137 | **0.244±0.012** |
| Deep LRP-5-1$^{\dagger}$ | - | - | 6590593 | 0.256±0.033 |
| Deep LRP-7-1$^{\dagger}$ | - | - | 11183233 | 0.223±0.008 |