[Reviews · NeurIPS 2020]

Review 1

Summary and Contributions: In this paper, the authors study the potential of graph neural networks in the problem of substructure counting. C1. Theoretical results are provided, suggesting the specific classes of graphs whose counting problems can or cannot be addressed by practical implementations of existing GNNs or k-WL variants. C2. Local Relational Pooling (LRP) is proposed to enhance the capability of counting substructures. C3. Empirical study on real-life and synthetic datasets suggests the proposed LRP outperforms existing GNNs in multiple substructure counting tasks.

Strengths: S1. This paper analyzes the limitation of existing GNNs in terms of substructure counting from both theoretical and empirical perspectives. S2. The authors propose a unique angle that enhances GNNs capability in substructure count by improving methods in local information aggregation. S3. The proposed LRP brings competitive performance in a few counting problems.

Weaknesses: W1. The authors may need to justify the concrete value of solving substructure counting problems by GNNs. Substructure counting is a problem that has been widely discussed in the domain of theory, database, and data mining. Existing combinatorial algorithms enhanced with improved system designs [1] enables accurate counting on large-scale graphs for simple substructures, such as triangle and stars discussed in this work. Compared with existing combinatorial method based solutions, what is the essential gain from using GNNs? W2. The design of LRP may also need both theoretical and empirical justification, compared with existing ideas. For example, in GraphSage, the idea of using sequential models (e.g., LSTM) to aggregate local information has been proposed. Given the same assumption that one is able to enumerate neighborhood permutation, it is non-trivial why the proposed LRP is better. The authors may provide more evidences to highlight the unique value in the proposed LRP, compared with existing solutions. W3. The presentation of the empirical results could be improved. In table 1, it is strange to report top and median performance, as the tested input graphs may not be aligned for different techniques. For example, the top 1 tested graph for solution A may not be the top 1 tested graph for solution B. When the input is different, it is confusing to compare their performance. Instead, it could be better to report average performance with mean and standard deviation for a same set of graphs. Reference [1] Suri, S et al. Counting Triangles and the Curse of the Last Reducer. WWW 2011.

Correctness: In general, the claims are correct, but the empirical study could be further improved, as suggested in the section of "weakness".

Clarity: Yes

Relation to Prior Work: In general, the related work is well addressed. Meanwhile, the authors may need to justify the unique value in LRP, compared with existing sequence model based neighborhood aggregation methods.

Reproducibility: Yes

Additional Feedback: Could the authors share some insights on why it is a good direction to improve GNN for substructure counting problems? For substructure counting, it is inevitable to deal with the subgraph isomorphism problem, which is NP-hard. In other words, it will be challenging to develop an efficient algorithm for exact counting or approximation with any approximation guarantee for arbitrary graphs and substructure patterns. For practical cases in real-life applications, existing combinatorial methods have made progress to improve scalability on large-scale data. To be concrete, what are the extra dimensions of improvement GNNs or deep learning methods could bring? ======After rebuttal===== Appreciate the authors' effort on addressing the questions. Indeed, the authors may provide a new perspective on "learning which substructure to count", which could be beyond the scope of existing combinatorial approaches. But still, without concrete evidences, it is a bit hard to envision the extra value of "learning which substructure to count". - Compared with existing GNNs which implement the concept of "counting" by pooling functions, it is unclear why accurate counting matters in prediction tasks. - Given the expressive power of GNNs, the impact of "learning which substructure to count" could be also limited. In sum, the value of "learning to count substructures" remains a bit vague. I keep the score unchanged.


Review 2

Summary and Contributions: This paper studies the expressive power of graph neural networks (GNNs) through their ability to subgraph-count and induced-subgraph-count. The paper examines MPNN, k-WL, and k-IGN. The theoretical results are dense and solid. Besides, authors derive a Local Relational Pooling (LRP) approach to count substructures in a graph, which is proven to be accurate empirically. Experiments further validate the proposed theories.

Strengths: The idea in this paper is novel. The proposed theories are promising. These can be used to guide future directions in developing GNN architectures. The empirical evaluation is sound. The content is relevant to the NeurIPS community.

Weaknesses: The proposed method LRP is computationally expensive and thus can be limited in practice. See more comments in additional feedback.

Correctness: Yes

Clarity: Yes

Relation to Prior Work: Yes

Reproducibility: Yes

Additional Feedback: 1. LRP is computationally expensive. The time complexity, O(n * (D!)^D^l * k^2), greatly depends on the degree of nodes. The paper considers LRP-1-4 (l=1), which makes the computation feasible but could limit the model's power. I am curious if the author managed to apply LRP with l>1 into any other datasets. 2. In (1), how to decide which subtensor C_k to use for node i? 3. In (2), where does the term $MLP(D_i)/|S^BFS_n|$ come from? This term is not shown in (1). 4. How the value bold in table 2? GIN+VN (77.07 +- 1.49) should be comparable to LRP-1-4 (ES) (77.19 +- 1.40), but GIN+VN is not bold. Thanks for the author response. I still think it's good addition for understanding GNN's capacity, and I'll keep my score.


Review 3

Summary and Contributions: In this paper the authors make a significant contribution towards understanding the expressive power of graph neural nets on a problem of great interest. A key concept of interest is the Weisfeiler-Lehman hierarchy, initially proposed by the expert on the GI problem Laszlo Babai. In this paper several neat results are proved. First, the authors prove MPNNs and 2nd order Invariant Graph Networks cannot count any connected induced subgraph of 3 or more nodes, but on the positive side they can count subgraphs that are star-shaped. Given the triangle rich structure of real-world networks this shows a severe limitation of these learning models. They also propose a new architecture that performs really well on real-data. [Update: I thank the authors for their feedback. My opinion remains the same, i.e., that this is a good paper that sheds light into a challenging problem.]

Strengths: - The paper is theoretically grounded. - The authors present a series of new important results, that advance the understanding we have of graph neural nets. - The architecture in section 5 performs really well on real-data - The experiments are well performed

Weaknesses: The writeup could be improved. Specifically, it was hard to parse some of the contributions in the main text, and some of the key results are stated without any intuitions behind the proofs.

Correctness: Yes, to the best of my knowledge the proofs are correct.

Clarity: Yes.

Relation to Prior Work: Yes.

Reproducibility: Yes

Additional Feedback: I understand that the page limit of 8 pages is limiting for a contribution like this one, but the authors could do a better job in revealing some more ideas behind their key results in the main text.


Review 4

Summary and Contributions: This paper propose a theoretical framework for studying the ability of GNNs to count attributed substructures based on both function approximation and graph discrimination.

Strengths: - The paper established that neither Message Passing Neural Networks (MPNNs) nor 2nd-order Invariant Graph Networks (2-IGNs) can count subgraphs more than 2 nodes. - The paper proposed that MPNNs and 2-IGNs can count subgraphs that are star-shaped. - The paper proposed a GNN model based on [41] called Local Relation Pooling.

Weaknesses: - The strength of the paper lies in the theoretical analysis, however, the findings are already known and not surprising, as it is well-known that GNNs are at most as expressive as WL. - Evaluation of LRP should be measured using standard benchmark graph datasets (MUTAG, Enzymes).

Correctness: Yes, the claims are correct.

Clarity: The paper is well presented, but should be checked for typos, some examples: - In line 71, "effectivel graphs"

Relation to Prior Work: Related work is discussed well.

Reproducibility: Yes

Additional Feedback: ============== After Rebuttal ================ Thanks for taking the time to write the rebuttal. I increased my score.

[Author Response · NeurIPS 2020]

We thank the reviewers for the constructive feedback, which confirms our contributions of laying a solid theoretical
foundation for studying the abilitiy for GNNs to count substructures, proving concrete results for MPNNs and IGNs,
and proposing the novel LRP model successful in substructure counting and real tasks. We address the concerns below.

**1. Motivation and Theory**

**R1** *The advantage of GNNs over traditional subgraph-counting algorithms?* Many challenging prediction tasks on
graph data can involve counting an **unknown** set of graph substructures, which calls for a data-driven solution that can
not only count substructures but also **discover** which substructures to count. GNNs have such potentials, provided that
they have enough expressive power to do so. Moreover, GNNs are known for integrating structural information with
node and edge features, which is desirable for counting **attributed** substructures efficiently.

**R4** *The theoretical results are not novel given existing results that GNNs are no more expressive than WL.* We respect-
fully disagree. While the reviewer is correct about prior work on the powers of MPNNs v.s. WL, our results precisely
establish substructure-counting (in)abilities of WL which were unknown before our work. These results then carry over
to MPNNs by building on prior work. Moreover, we also prove results for IGNs, which are not within the WL hierarchy.

**2. The Local Relational Pooling model**

**R1** and **R2** *Time-complexity of LRP.* While enumerating node permutations may sound prohibitive, LRP can be imple-
mented practically, thanks to 1) only enumerating each local egonet; 2) only considering BFS-consistent permutations;
3) tensor cropping; 4) pre-computing node/edge indices for permutations. Its efficiency in practice is shown in Table 1.

**R1** *How does GraphSAGE compare with LRP?* Indeed, the expressiveness of GraphSAGE
deserves a separate analysis, especially when using multi-hop neighborhood. If we allow
GraphSAGE to enumerate all permutations of the neighborhood, as the reviewer suggested,
it becomes similar in spirit to LRP, except for aggregating neighborhood information as a
sequence rather than a tensor, which results in advantages of LRP in exploiting edge features and high-order structures:
1) While the original GraphSAGE does not consider edge features, even if we can do so via augmenting the node features
by applying an invariant function to the features of its immediate edges (e.g. summing or averaging), GraphSAGE
cannot distinguish the pairs of graphs shown on the right, while LRP-1-3 can; 2) GraphSAGE cannot distinguish the
pair of 12-circular graphs $C_{12}(1,3)$ and $C_{12}(1,5)$ (see [1]), no matter the hop-size being used, because $\forall k$, the $k$-hop
neighborhood of every node in the two graphs has the same size. This means GraphSAGE cannot count squares, while
LRP-2-4 can. Further, Table 3 shows the performance on the synthetic tasks of GraphSAGE + LSTM on full 1-hop
neighborhood - it can count stars but not triangles, consistent with the limitation of information in 1-hop neighborhood.

**R2** *How to decide which subtensor $C_k$ to use for node $i$?* Let $G_{i,l}^{\mathsf{ego}}$ denote the egonet of depth $l$ centered at node $i$. For
every BFS-consistent permutation, $\pi$, of the nodes in $G_{i,l}^{\mathsf{ego}}$, $C_k(\pi \circ \mathbf{B}_{i,l}^{\mathsf{ego}}) \in \mathbb{R}^{k \times k \times d}$ gives the tensor representation of
the subgraph induced by the first $k$ nodes under this permutation. On the choice of $k$ - we set $k$ to be an upper bound on
the size of the substructures of interest.

**R2** *Where does the term* $\mathtt{MLP}(D_i)/|S_n^{BFS}|$ *come from?* If this term were 1 or $1/|S_n^{BFS}|$, we would get sum or mean
pooling over all BFS-consistent permutations of the egonet, respectively. However, adding irrelevant edges to node $i$
affects both the total number and fraction of permutations in which a substructure of interest appears, and so neither
summing nor averaging over all permutations is fully desirable. Hence, we introduce $\mathtt{MLP}(D_i)$ to learn to correct this bias
as a function of the degree of node $i$. Thus, we can learn an invariant function over permutations that extends summation
and averaging. From the literature of GNNs, this can be also seen as a generalization of the degree-normalization in
GCNs. When $l > 1$, we generalize from $\mathtt{MLP}(D_i)$ to an MLP over the list of degrees of all nodes in the permutation.

**3. Experiments**

**R1** *Regarding what "top and median performances" means.* Sorry about not having made it clearer, but here "top" and
"median" are with respect to five random seeds in training, and the errors are indeed averaged over all test graphs. The
reason we report the top performance is because we are more interested in expressive power than training, and a good
top performance suffices to indicates good expressive power.

**R4** and **R2** *Experiments on more standard datasets; How about LRP with $l > 1$?* Additional results on ZINC and
MUTAG are shown below. When $l > 1$, each cropped subtensor contains one node from every depth level $\leq l$.
LRP-7-8 almost matches the best performance (by GateGCN-E-PE, which augments node features with top Laplacian
eigenvectors) on ZINC benchmarked in [2]. LRP-1-4 also surpasses GIN and 3WLGNN ([3]) on MUTAG.

Table 1: Results for ZINC. †: reported in [2].

| Model | Train MAE | Test MAE | Time / Epoch |
|---|---|---|---|
| GraphSAGE† | $0.081 \pm 0.009$ | $0.398 \pm 0.002$ | 16.61s |
| GIN† | $0.319 \pm 0.015$ | $0.387 \pm 0.015$ | 2.29s |
| MoNet† | $0.093 \pm 0.014$ | $0.292 \pm 0.006$ | 10.82s |
| GatedGCN-E-PE† | $0.067 \pm 0.019$ | $0.214 \pm 0.013$ | 10.70s |
| GatedGCN-E† | $0.074 \pm 0.016$ | $0.282 \pm 0.015$ | 20.50s |
| 3WLGNN† | $0.140 \pm 0.044$ | $0.256 \pm 0.054$ | 334.69s |
| LRP-7-8 | $0.028 \pm 0.004$ | $0.223 \pm 0.008$ | 72s |
| LRP-5-6 | $0.020 \pm 0.006$ | $0.256 \pm 0.033$ | 42s |

Table 2: Results for MUTAG. †: reported in [3].

| Model | Test Acc (%) |
|---|---|
| FGSD† | 92.12 |
| GIN† | $89.4 \pm 5.6$ |
| 3WLGNN† | $90.6 \pm 9.7$ |
| LRP-1-4 | $91.0 \pm 6.4$ |

Table 3: Top & median loss on synthetic tasks.

| Model | 3-Star | Triangle |
|---|---|---|
| GraphSAGE | 2.4E-10 / 2.0E-5 | **1.3E-1 / 1.5E-1** |
| LRP-1-4 | 1.1E-5 / 3.8E-5 | **2.8E-5 / 4.8E-5** |

[1] Weisstein, "Quartic Graph." [2] Dwivedi et al., "Benchmarking GNNs." [3] Maron et al., "Provably Powerful Graph Networks."


[Meta-Review · NeurIPS 2020]

There was consensus from the referees that this paper provides interesting theory while also shedding light on what graph neural networks might be learning. For that reason, the recommendation is to accept. That being said, there is one important point that remains unclear. There are a number of fast and easy algorithms that one could use to count substructures / graphlets / motifs, so why not just use these algorithms to produce node features? (For example, see “Efficiently Counting Vertex Orbits of All 5-vertex Subgraphs, by EVOKE” by Pashanasangi and Seshadhri and references therein for a large body of literature on the topic of counting subgraph patterns.) The author response addresses this concern in part by arguing that the GNNs could learn the right attributed subgraph patterns. That seems like much better motivation, and I would encourage emphasizing that point in the camera version. To augment this, the experiments on the “Real tasks” could include a pre-processing step, where subgraph participation counts are computed and then introduced as node features. If this substantially changes the performance, then there would be additional substantiation for the story in the paper.